

# Evolution of the chemical fingerprint of biomass burning organic aerosol during aging

Amelie Bertrand[1,2,*], Giulia Stefenelli[3], Coty N. Jen[4], Simone M. Pieber[3], Emily A. Bruns[3], Brice Temime-Roussel[1], Jay G. Slowik[3], Allen H. Goldstein[4], Imad El Haddad[3], Urs Baltensperger[3], André S.H. Prévôt[3], Henri Wortham[1] and Nicolas Marchand[1]

[1]Aix Marseille Univ, CNRS, LCE, Marseille France
[2]Agence de l'environnement et de la Maîtrise de l'Energie, 20, avenue du Grösillé – BP 90406 49004 Angers cedex 01 France
[3]Laboratory of Atmospheric Chemistry, Paul Scherrer Institute, 5232, Villigen, Switzerland
Department of Environmental Sciences, Policy, and Management, University of California at Berkeley, California, United States
[*]Now at Laboratory of Atmospheric Chemistry, Paul Scherrer Institute, 5232, Villigen, Switzerland

*Correspondence to*: Nicolas Marchand (nicolas.marchand@univ-amu.fr)



**Abstract.** A Thermal Desorption Aerosol Gas Chromatograph coupled to a High Resolution – Time of Flight – Aerosol Mass Spectrometer (TAG-AMS) was connected to an atmospheric chamber for the molecular characterization of the evolution organic aerosol (OA) emitted by woodstoves appliances for residential heating. Two logwood stoves (old and modern) and one pellet stove were operated under typical conditions. Emissions were aged during a time equivalent to 5 hours of atmospheric aging. 5 to 7 samples were collected and analyzed with TAG-AMS during each experiment. We detect and quantify over 70 compounds, including levoglucosan and nitrocatechols. We calculate the emission emissions factor (EF) of these tracers in the primary emissions and highlight the influence of the combustion efficiency on these emissions. Smoldering combustion contribute to higher EF and a more complex composition. We also demonstrate the effect of the atmospheric aging on the chemical fingerprint. The tracers are sorted into 3 categories according to the evolution of their concentration: primary compounds, non-conventional primary compounds, and secondary compounds. For each we provide a quantitative overview of their contribution to the OA mass at different times of the photo-oxidative process.



# 1 Introduction

Organic matter represents a major fraction (20 – 90 %) of particulate matter (PM) (Kanakidou et al., 2005). Organic PM is a complex mixture made up of tens of thousands of compounds (Goldstein and Galbally, 2007), with some of them established to be carcinogenic (Yu, 2002), (Yang et al., 2010). Identifying and quantifying their contribution to the organic PM mass is key in order to resolve its origins and impacts on human health and climate.

Extensive characterization of the molecular composition of primary organic aerosol (POA) emissions has offered the identification of useful tracers of specific sources. For example, biomass burning emissions (Simoneit et al., 1993; Fine et al., 2001 ; Fine et al., 2002 ; 2004 ; Nolte et al., 2001 ; Schauer et al., 2001; Schmidl et al., 2008), vehicular emissions (Rogge et al. 1993; 1993b; Fraser et al., 1999 ; Schauer et al., 2002, El Haddad et al., 2009), and cooking emissions (Hildemann et al., 1991; Nolte et al., 1999; Schauer et al., 1999) have been broadly characterized. In biomass smoke, compounds derived from the pyrolysis of cellulose and lignin are often reported. These include levoglucosan, a sugar anhydride compound and by-product of the thermal degradation of cellulose, and methoxyphenols, by-products of the thermal-degradation of lignin. Their relative amount can vary with the type of fuel (hardwood, softwood, or herbaceous types) (Schmidl et al., 2008b; Schauer et al., 2001), the type of fire (open fire, fire places, wood stove) (Fine et al., 2002; 2004), or even the sampling set-up (in an experimental stack, in a dilution tunnel, or ambient) (Nussbaumer et al., 2010). The full characterization of these emissions is of particular interest for source apportionment of ambient PM using molecular markers. Levoglucosan, for instance, is a commonly emitted tracer of biomass burning. Its ubiquity and abundance (Waked et al., 2014; Bonvalot et al., 2016; Maenhaut et al., 2016) have been used to demonstrate the significant contribution of biomass burning to the total organic aerosol source globally (Robinson et al., 2006; Gelencsér et al., 2007; Puxbaum et al., 2007; Stone et al., 2010; Crippa et al., 2013).

The concentration of organic aerosol (OA) particle mass has been documented to increase up to 7 times during photochemical aging (Grieshop et al., 2009; Heringa et al., 2011; Ortega et al., 2013; Bruns et al., 2015; Tiitta et al., 2016), however, the chemical composition of this secondary organic aerosol (SOA) produced remains uncertain. Multiple studies investigated the oxidation of specific gas-phase precursors commonly emitted by biomass burning, namely methoxyphenols (Net et al., 2011; Lauraguais et al., 2012; Yee et al., 2013; Lauraguais et al., 2014), but few have specifically addressed the aging of specific biomass burning tracers in the particle phase (Hennigan et al., 2010; Kessler et al., 2010; Lai et al., 2014), and so far, only Fortenberry et al. (2017) have attempted characterizing the aged chemical fingerprint of biomass burning emissions at the molecular level by means of a Thermal Desorption Aerosol Gas Chromatograph (TAG) connected to a Potential Aerosol Mass (PAM) flow reactor. This is especially important in the context of source apportionment studies which assume that chemical profiles are constant with time.

In a previous publication, we investigated the POA emissions and SOA production potential generated by three woodstove appliances (two logwood stoves and one pellet stove) used for residential heating (Bertrand et al., 2017) using a HR-ToF-AMS. Here, we provide a comprehensive study, including the evolution of the molecular level composition of the



emissions during a period equivalent to 5 hours of atmospheric aging. The experiments were conducted using the atmospheric chamber of the Paul Scherrer Institute (PSI, Villigen, Switzerland). The data were obtained by means of a TAG-AMS (Aerodyne Research Inc.). We determine the emission factors (EF) and emission profiles of biomass burning tracers. In a first part, we derive the effect of combustion conditions on these EFs and their contribution to the POA mass, and in a
second approach we determine the effect of the atmospheric aging on their contribution to the total OA mass.

## 2 Materials and Methods

### 2.1 Set up

The full set-up and protocol was previously fully described in (Bertrand et al., 2017). The atmospheric chamber is a 5.5 m$^3$ Teflon chamber with a set of $40 \times 100$ W UV lights (310 – 430 nm (Platt et al., 2013)) to initiate photo-chemistry (Figure 1).
Whereas most studies conduct their experiment at ambient temperature, here the chamber was set to 2 °C in order to simulate wintertime conditions. Relative humidity (RH) was kept at 50 %. Primary and aged emissions were characterized using a suite of instrumentation. This included a TAG-AMS (Aerodyne Research Inc.) for the online speciation of the OA particle mass, an Aethalometer (Magee Scientific Aethalometer model AE33) (Drinovec et al., 2015) for the quantification of the equivalent black carbon (BC) concentration, a HR-ToF-AMS (Aerodyne Research Inc.) for the bulk-condensed chemical
composition of the non-refractory fraction of the aerosol, and a Proton Transfer Reaction – Time of Flight - Mass Spectrometer (PTR-ToF-MS 8000, Ionicon Analytik) for the monitoring of volatile organic compounds (VOCs). The PTR-ToF-MS was operated under standard conditions, i.e. ion drift pressure at 2.2 mbar and drift field intensity at 125 Td. The HR-ToF-AMS was equipped with a PM$_{2.5}$ aerodynamic lens and operated under standard conditions, i.e. the temperature of the vaporizer set at 600 °C, and the electron ionization (EI) at 70 eV.
Emissions were generated by three different wood burning appliances. Stove A was a logwood stove fabricated before 2002, stove B a logwood fabricated in 2010, and stove C a pellet stove from 2010. For each stove, experiments were replicated 3 or 4 times (Table 1). The logwood stoves were loaded with 2 – 3 kg of beech wood as logs and kindling wood, and with moisture content between 10 and 12 %. The pellet stove was fueled with commercial pellets, considered of premium quality (European Standards, 2012), composed of a mixture of pine and spruce wood, and with a moisture content
of 7.7 % and a density of 600 kg m$^{-3}$. The stoves were connected from the chimney to the chamber via heated (140 °C) Silico-steel lines.

### 2.2 Experimental protocol

Following ignition and prior to injection in the chamber, the emissions were diluted by a factor of 10 through an ejector dilutor (DI-1000, Dekati Ltd.). After injection and prior to aging, the mass concentration of OA in the chamber ranged
between 10 and 177 μg m$^{-3}$ (Table 1) and can thus be considered representative of ambient to plume-like conditions. The concentration of NO$_x$ ranged from 50 to 255 ppb. Primary emissions were left static for approximately 30 minutes for





stabilization and full characterization. Then, 1 µL of butanol-D9 (butanol-D9, 98%, Cambridge Isotope Laboratories) was injected. Butanol-D9 is a commonly used tracer for hydroxyl radicals in atmospheric chamber studies (Platt et al., 2013; Bruns et al., 2015; Klein et al., 2016). Its fragment at *m/z* 66.126 ($[C_4D_9]^+$) can be easily monitored by means of a PTR-ToF-MS (Barmet et al., 2012). Nitrous acid (HONO) was injected continuously at a flow rate of 1 L min$^{-1}$. HONO dissociates

under UV lights (λ < 400 nm) to form OH. Here, to integrate the dilution factor owing to the continuous injection of HONO in the chamber, we retrieved the OH exposure based on the differential reactivity of butanol-D9 and naphthalene ($[C_{10}H_8]H^+$, *m/z* 129.070). The photo-oxidation of the emissions lasted approximately for four hours. We calculated an integrated OH exposure at the end of our experiments which ranged between $5 \times 10^6$ molecules cm$^{-3}$ h and $8 \times 10^6$ molecules cm$^{-3}$ h. This translates to 5 to 8 hours of atmospheric aging, respectively, with a typical average daytime OH concentration of $1 \times 10^6$ OH

molecules cm$^{-3}$.

**2.3 Operation of the TAG-AMS**

The TAG-AMS (Figure 1) permits the on-line collection and analysis of the OA particles at the molecular level with a time resolution of less than an hour. The instrument presented here is a modified version of the original TAG system (Williams et al., 2006). The TAG is coupled to a modified HR-ToF-AMS (DeCarlo et al., 2006) equipped with a quadrupole high pass-

filter (Tofwerk) and opened at the rear flange to accommodate a transfer line (Williams et al., 2014). The quadrupole allows for the discrimination of all fragments below *m/z* < 10. This avoids the saturation of the detector by helium, used as the carrier gas in the TAG system. The transfer line connects the Gas Chromatography (GC) column of the TAG to the bottom of the ion cage inside the AMS. It consists of a 15" length × 1/32" OD coated stainless steel capillary which travels through a heated copper rod. The transfer line is maintained at all times at 300 °C. This TAG-AMS was also implemented with an

online derivatization system for the analysis of more polar compounds (Isaacman et al., 2014).

Emissions are sampled and passed through a parallel plate charcoal denuder to remove the trace gases. The aerosol is then humidified (65 – 95 % RH) to enhance the collection efficiency onto the Collection Thermal Desorption (CTD) cell. The CTD cell is kept at ambient temperature during sampling and then heated progressively (in 4 minutes) to 280 °C for the measurement. Analytes are thermally desorbed into a helium flow enriched with the derivatization agent N-methyl-N-

trimethylsilyl trifluoroacetamide (MSTFA, Sigma Aldrich). This derivatization flow is then routed to the cell in parallel of a pure helium flow (purge flow). The derivatization and purge flow were set at 40 and 15 cm$^3$ min$^{-1}$, respectively, so as to ensure complete derivatization of the more polar compounds. The organic material is transferred onto a focusing trap (15 m × 0.53 mm ID × 5 µm df MXT-1 Restek column). During the transfer, the column is kept at 40 °C. Excess MSTFA is flushed out of the system in a subsequent purging step for approximately 4 minutes. In addition, compounds with a higher

volatility than MSTFA are also flushed out. However, this does not affect the analytes of interest studied here. The separation of the analytes is achieved using a 15 m long x 0.18 mm diameter ValcoBond-5 fastGC capillary column. The initial temperature of the column is 45 °C and is increased by 36 °C min$^{-1}$ to 100 °C, then by 30 °C min$^{-1}$ to 310 °C, and held for 5 minutes. We show chromatograms in Figure S1 of the supplementary informations (SI). Peak fitting and integration are





achieved using the TERN (version 1.0) (Isaacman-VanWertz et al., 2017) data analysis toolkit in Igor Pro 6.3 (Wave Metrics).

For optimum results, the TAG-AMS should sample air at 9 L min$^{-1}$. In order to make full use of the TAG-AMS without rapidly depleting the volume of the chamber, the sampling flow rate was restricted to 2 L min$^{-1}$. An additional line

which sampled outside air filtered through a high-efficiency particulate air (HEPA) filter was installed to make up for the missing flow rate.

Five to seven samples were collected in each experiment. The first sample was systematically collected before photo-oxidation of the emissions (POA/fresh sample). Sampling lasted between 5 and 25 minutes, depending on the OA concentration measured in the chamber by the HR-ToF-AMS. To compensate for the loss of materials to the walls during

aging, the TAG-AMS sampling time was typically increased with each new sampling (Figure 2).

Identification of the compounds for which the authentic standards were available (Table S1) was performed based on their retention times and EI mass spectra. Identification of the compounds for which the authentic standards were not available was performed by matching their EI mass spectrum to the Wiley and National Institute of Standards and Technology (NIST) mass spectral libraries. The identification of the compounds was also supported by 2D-GC analysis in

both EI and vacuum ultra violet (VUV) ionization carried out on one pair of offline samples (quartz fiber filters) collected before and at the end of the aging phase, as illustrated in Figure 2. Details of the 2D-GC analytical procedure are provided in the SI (Figure S2). Overall, we identified 71 compounds. They include the anhydrosugars levoglucosan, mannosan, and galactosan, the methoxyphenols (substituted guaiacols and substituted syringols, details in Section 3.2.2), as well as a variety of polycyclic aromatic hydrocarbon (PAHs) (including methylated-PAHs and oxygenated PAHs), alkanes (C$_{18}$ to C$_{27}$), fatty

acids (saturated and unsaturated), and a variety of tracers found almost exclusively in the secondary emissions (see Section 3.2.3).

For quantification, a deuterated internal standard mixture in acetonitrile was added prior to analysis of each sample in the CTD cell via an automatic injection system developed by Isaacman et al., (2011). The standard mixture included adipic acid-D10, phthalic acid-D4, eicosane-D42, and tetracosane-D50 (Sigma Aldrich). A 5-point mass calibration curve of

the authentic standards was achieved before and after the campaign. One point was rerun all along the campaign to check the calibration. The response varied by < 10 %. Compounds for which the authentic standard was not available were quantified using an appropriate surrogate (Table S1). Detection limits are presented in Table S2.

## 2.4 Particle wall loss corrections (pWLC) and emission factor (EF) calculations

The mass concentrations were corrected for particle wall loss following the method described in Weitkamp et al. (2007) and

Hildebrandt et al. (2009). Briefly, the first order decay of an inert tracer, here eBC, was used to estimate the fraction of condensed material lost to the wall. The corrected mass concentration for particles wall losses $C_{pWLC}$ of a species was calculated following Equation 1:



$$C_{pWLC}(t) = C(t) + \int_0^t k_{wall/p}(t).C(t).dt \tag{1}$$

where $C$ is the concentration of a species measured by the TAG-AMS or HR-ToF-AMS, $t$ is the time (in min) and $k_{wall/p}$ is the eBC wall loss rate constant (in min⁻¹). Unless stated otherwise, all concentrations and contributions of a marker to OA reported below are corrected for particle wall loss.

This method assumes that the condensable material partitions only to the suspended particles and vapor wall losses are considered negligible. The corrected OA mass concentration and that of the individual markers thus correspond to the lowest estimate possible of the total mass formed. Unlike particle wall loss, vapor wall loss cannot be easily constrained by external measurements. To estimate the loss rate, one would need to conduct a thorough investigation of several key parameters (e.g., each compound's saturation vapor concentration, particle mass accommodation coefficient, or equivalent organic mass concentration at the wall) which are not yet properly constrained within the existing literature. Investigation into vapor wall loss is beyond the scope of this paper but is addressed in Bertrand et al (2018b, in preparation). Concentrations reported here are not corrected for vapor wall loss.

EF were calculated based the method by Andreae and Merlet (2001) (Equation 2). The equation relates the mass emitted of a pollutant P to that of the amount of fuel burnt.

$$EF_p = \frac{\Delta P}{\Delta C_{CO_2} + \Delta C_{CO} + \Delta C_{THC} + \Delta C_{OC} + \Delta C_{BC}}. W_c \tag{2}$$

Here Δ refers to the concentrations of the species in the atmospheric chamber after emission and stabilization, and corrected for their background levels in the atmospheric chamber before injection. C relates to the carbon masses of $CO_2$, CO, THC (including methane), organic carbon (OC), and black carbon (eBC). OC was inferred from the OM/OC ratio determined with the high resolution AMS analysis (Canagaratna et al., 2015). The carbon content ($W_c$) of the beech logs and the pellets was determined in the laboratory by analyzing sawdust of the wood samples with an Elemental Analyzer (Flash HT, Thermo-Fisher) and was 46 % ± 1 % ($n = 8$, w/w) for both types of wood (Bertrand et al., 2017).

## 3 Results

### 3.1 Effect of burning conditions on the POA chemical composition

EFs of the primary emissions are reported in Table S3 of the SI for all experiments. Figure 3 shows the relative chemical composition of the identified fraction of the POA mass for each stove (averaged over the replicates). We identify between 26 - 85 % of the POA mass concentration. Previous studies have revealed a relationship between the modified combustion efficiency (MCE) and characteristics of the mass spectral signature of the OA emissions (Jolleys et al., 2014; Bertrand et al., 2017). The MCE values for the logwood stoves (Stove A and B) range between 0.80 - 0.91 (Table 1), indicating that the



combustion in these stoves is typically smoldering (MCE < 0.9), but also highly variable). However, the pellet stove (Stove C) shows little variability and produces a flaming type of combustion. The MCE values for the experiments conducted with this stove are 0.97. As illustrated in Figure 4, a trend exists between the overall identified fraction and the MCE. We typically identify a lower fraction of the POA mass concentration in emissions from smoldering type of combustions. In the

next Section we further investigate the influence of the burning conditions on specific compounds.

### 3.1.1 Levoglucosan and other anhydrosugars

The EFs of levoglucosan are in the range 26 - 249 mg kg$^{-1}$ of fuel or 13 - 45 % of the POA mass concentration. Levoglucosan is emitted along with its two isomers, mannosan, and galactosan. The EFs of mannosan are 4 - 20 mg kg$^{-1}$ and represent an average fraction of the POA mass concentration of < 4 % ± 2 % ($n = 11$). The EFs of galactosan are 0.5 - 10 mg

kg$^{-1}$ and contribute < 1 % to the POA mass concentration. The percentage of anhydrosugars in the POA mixture is on average higher in the case of the pellet stove (Stove C) experiments (42 % ± 9 %, $n = 3$) compared to Stove A (27 % ± 11 %, $n = 4$) and Stove B (32 % ± 13 %, $n = 4$). For comparison, previous studies detailing the chemical composition of BBOA have typically reported lower contributions of anhydrosugars to the POA mass concentration (7 %, $n = 6$) (Fine et al., 2001), (7 %, $n = 6$) (Fine et al., 2002), (18 %, $n = 7$) (Fine et al., 2004), (24 %, $n = 3$) (Schauer et al., 2001). The conditions and

methods with which they sampled the emissions were however different (on quartz fibers filters, directly in the stack or at ambient temperature), and have often been reported to cause bias in the measurements due to positive and/or negative artefacts (Turpin et al., 1994; 2000).

Figure 4 shows the EF of levoglucosan and its contribution to the POA as a function of the MCE. The emission of levoglucosan correlates well with the MCE ($R^2 = 0.61$, $n = 11$ shown in Figure 6). Overall, emissions increase with lower

MCE while the actual contribution of levoglucosan to the POA mass concentration decreases with lower MCE. We consider therefore that while smoldering combustions increase the emission of levoglucosan, this process also results in large emissions of a variety of other compounds along with the anhydrosugars; the chemical composition of the resulting organic aerosol is thus more complex and the identified fraction of the OA is lower. This substantial difference in the contribution (from 40 % – 50 % at the highest MCE, down to 15 % at the lowest MCE) is however never considered in source profiles for

source apportionment studies.

### 3.1.2 Methoxyphenols

The methoxyphenols account for a predominant fraction of the POA, representing between 8 % and 27 % of the POA. They include substituted guaiacol compounds such as vanillin, acetovanillone, vanillic acid, 3-guaiacylpropanol, coniferyl aldehyde as well as syringol and substituted syringols such as iso-eugenol, syringaldehyde, acetosyringone, syringyl acetone,

propionyl syringol, syringic acid, methyl syringol, and synapyl aldehyde. The total EF of the methoxyphenols is between 7 and 174 mg kg$^{-1}$. In the case of the logwood stoves (stove A and B), the methoxyphenols contribute on average 17 ± 6 % of the total POA mass concentration. This fraction decreases to 9 % ± 0.7 % ($n = 3$) for the experiments conducted with the



pellet stove (stove C). Syringyl acetone and vanillin are the most abundant compounds in their respective category with EFs ranging from 0.4 to 80 mg kg$^{-1}$ and from 2 to 7 mg kg$^{-1}$, respectively.

As demonstrated in Fine et al. (2001; 2002; Fine et al., 2004), emissions from softwood combustion contain a larger proportion of substituted guaiacols than of substituted syringols, while emissions from hardwood combustion contain a larger amount of substituted syringols than of guaiacols. Our results are consistent with these data. The emissions of the logwood stoves A and B (hardwood) contain up to 9 times higher amounts of substituted syringols than of substituted guaiacols. The emissions by the pellet stove are more balanced, with a ratio of substituted syringols to substituted guaiacols ratio of less than 2.

As for levoglucosan, we observe a significant increase of the absolute EFs of methoxyphenols with a decrease of the MCE (Figure 4), although not as strong as the POA EF increase. For the hardwood combustion (Stoves A and B), this results in substantial increase of the overall contribution of the methoxyphenols to the POA mass concentration with the MCE. At smoldering conditions, the average contribution of methoxyphenols to the POA mass concentration is 15 % ± 4 % while it averages at 22 % at flaming conditions.

### 3.1.3 Other compounds

On average, the contributions of the other identified compounds (PAHs, alkanes, fatty acids) to the total POA mass concentration are higher in the case of pure flaming combustion experiments (Stove C, MCE = 0.97). The relative contributions of the PAHs to the POA mass concentration reach 1.5 ± 1.1 %, 2.2 ± 1.2 %, and 1.9 % ± 0.3 %, respectively for stove A, stove B, and stove C. Phenanthrene and fluoranthrene are the most dominant as they represent over a third of the total PAH contributions. The EF of the individual PAHs vary between 3 μg kg$^{-1}$ and 4 mg kg$^{-1}$, with combined amounts between 2 and 14 mg kg$^{-1}$. The alkanes represent 0.21 ± 0.01 %, 0.36 ± 0.02 %, and 0.64 % ± 0.03 % of the total POA mass concentration, respectively for stove A, B, and C. The total EFs vary between 0.5 and 2 mg kg$^{-1}$. Four series of fatty acids including the saturated acids palmitic acid and stearic acid, and the unsaturated acids palmitoleic acid and oleic acid, were identified. They contribute 0.33 ± 0.08 %, 0.32 ± 0.10 %, and 0.87 % ± 0.31 % to the total POA mass for stove A, stove B, and stove C, respectively. Their total EFs are similar to the alkanes. Overall, the sum of PAHs, alkanes, and fatty acids accounts for less than 3 % of the total POA mass.

### 3.2 Effect of atmospheric aging on the chemical fingerprint of OA

As illustrated in Figure 5, the molecular fingerprint of the BBOA undergoes important modifications during aging. Considering that such fast modifications can generate significant biases in our ability to properly apportion the biomass burning source in the ambient atmosphere, quantitative data on the chemical changes of the molecular signature are required. Here, we report EFs of all the quantified molecular markers at an integrated OH exposure of $5 \times 10^{6}$ molecules cm$^{-3}$ h in Table S3. The contributions of the most relevant markers to the total OA mass concentration at different stages of the photo-oxidative process are also reported in Table 2. A more exhaustive table is also available in the SI (Table 4). While TAG-



AMS samples are collected and analyzed approximately at the same time (after lights on) between the experiments, the integrated OH exposure however is not homogeneous. Therefore, data are divided into seven bins of different integrated OH exposure intervals (Figure 6) (from 0 to $9 \times 10^6$ molecules cm$^{-3}$ hour). The intervals are set to best reflect the evolution of the compounds.

Figure 6 further illustrates the quantitative modification of the chemical fingerprint during the aging process for one experiment (Experiment 2). In this example, while the quantified markers represent 32 % of the POA, only 4.5 % of the total OA is identified by the time the experiment reaches an integrated OH exposure of $5 \times 10^6$ molecules cm$^{-3}$ h. This result can be generalized since < 10 % of the POA is quantified at the molecular level after the photo-oxidative process. This can be explained by the nature of the emissions present at this stage of the experiment. Due to the oxidative mechanisms taking

place, aged OA emissions contain a large amount of high molecular weight, low volatility, and highly oxygenated compounds (Donahue et al., 2011; Ng et al., 2011) with multiple isomers. This is reflected in the modification of the O:C ratio (from 0.45 to 0.65). Many of these more oxidized and highly functionalized compounds cannot be easily identified by the methods used here either because they do not elute from the TAG-AMS, or because their mass spectra are not present in the NIST libraries.

Based on the evolution of their absolute concentrations and their enhancement ratios (ER), (defined as the ratio between pWLC concentration measured in the particle phase at a time *t* and the concentration in the particle phase measured at $t_0$ or before lights on), the compounds are classified within 3 categories:

-      primary compounds (ER < 1)
-      non-conventional primary compounds (1 < ER < 4)
-      secondary compounds (ER > 10)

       Figure 7 shows the ERs of all compounds in their respective category. Primary compounds are compounds emitted during the combustion and whose concentration in the atmospheric chamber is found to decrease during aging. The non-conventional primary compounds are compounds emitted along other conventional primary compounds during combustion but whose concentration is found to increase during aging. Note that as the concentrations of the non-conventional primary

compounds increase more slowly than OA, their relative contributions to OA slightly decrease during the aging phase. The secondary compounds are compounds principally found after lights on. In certain experiments, these were also detected in minor amounts in the primary emissions. We distinguish them from the non-conventional primary compounds by the relative amount formed during aging (over ten times the concentration measured in the primary emissions).

### 3.2.1 Primary compounds

The identified primary compounds initially represent 48 % of the POA mass concentration but < 8 % after 4 - 6 hours of atmospheric aging (bin 4). Levoglucosan and syringyl acetone are the most abundant compounds of this category. Other compounds include the isomers of levoglucosan, mannosan, and galactosan, as well as the PAHs, alkanes, and the following





methoxyphenols: 3-guaiacylpropanol, conyferyl aldehyde, syringol, iso-eugenol, syringaldehyde, acetosyringone, propionyl syringol, methyl syringol, and synapyl aldehyde.

Before lights on, levoglucosan contributes on average 30 % of the total OA mass concentration. Its contribution decreases to 4.5 % after approximately 4 – 6 hours of atmospheric aging, i.e., by a factor of 6.6. Within the first hour, its contribution to OA is decreased by a factor of 3 (Figure 8). The mean ER of levoglucosan is 0.6 (0.4 – 0.8). The decrease of mannosan is on a similar scale, contributing < 4 % of the total OA before lights on and then decreasing to 0.5 % after aging, i.e., decreasing by a factor of 7. Its mean ER is also 0.6 (0.4 – 0.8).

Before lights on, syringyl acetone contributes 4.7 % of the total OA mass concentration. Its contribution decreases to 0.2 % after aging (a decrease by a factor of 23.5 %). Within 1 hour of atmospheric aging, its contribution to OA has decreased by a factor of 5. Its mean ER is 0.3 (0.1 – 0.7). The contribution of the other identified methoxyphenols to the total OA mass concentration decrease by an average factor of 17 after 4 – 6 hours of atmospheric aging (as high as a factor of 43 for synapyl aldehyde and as low as 6 for syringaldehyde).

The PAHs and alkanes show a more moderate decrease. Their contribution to the total OA mass concentration decreases by approximately 2 to 5 times after 4 – 6 hours of atmospheric aging. The high molecular weight PAHs are an exception. Benzofluoranthene and other five-rings PAHs are observed to decrease by a factor > 10 after 4 – 6 hours of atmospheric aging.

The rate at which these compounds decrease is of particular interest. Many serve as molecular markers in source apportionment studies and are as such implicitly assumed to not oxidize or volatilize. If they are not stable these markers will introduce a bias in the results and lead to an underestimation of the contribution of biomass burning in favor of other sources including the secondary fraction. Previous studies have examined the effective rate constant for the reaction of these compounds with OH (Lambe et al., 2009b; Hennigan et al., 2010; Kessler et al., 2010; Hennigan et al., 2011; Lai et al., 2014) at ambient temperature, and the rate at which levoglucosan decreases here is consistent with their results (at 2°C). However, we note that our results ignore the semi-volatile nature of many of these compounds and likely wall losses. As demonstrated by (May et al., 2012; Zhang et al., 2015) ignoring wall losses can result in an underestimation of the compound concentration measured in the chamber. Therefore, the $k_{OH}$ rate constant derived from these results must be regarded with caution. This is addressed in further detail in Bertrand et al. (2018b, in preparation).

### 3.2.2 Non-conventional primary compounds

This category includes mainly methoxyphenols such as vanillic acid, vanillin, acetovanillone, and syringic acid, and some less known compounds: tyrosol and pyrogallol. Their contribution in the primary emissions is not significant (on average less than 3 % of the POA mass concentration in total). Unlike the other primary compounds emitted during combustion, their pWLC concentration is found to increase during aging. Fortenberry et al. (2017) observed a similar trend for vanillin. As most of these compounds are detected both in the gas and particle phase of the emissions (Bruns et al., 2017; Schauer et al., 2001), with the smaller fraction detected in the particle phase of the emissions, it is possible that their increase is partly a





result of the partitioning effect. A significant increase in the OA mass after lights on as observed in these experiments could drive a larger fraction into the particle phase. However, it is likely that this effect would be reduced by the vapor wall loss also occurring, which has been demonstrated to be significant by other authors (Zhang et al., 2015; La et al., 2016; Trump et al., 2016; Ye et al., 2016). Therefore, the other most sensible hypothesis is that these compounds are formed during the oxidation process. Several experiments have been conducted on the heterogeneous and gas-phase reactivity of methoxyphenols with atmospheric oxidants (OH, NO$_3$) (Net et al., 2011; Lauraguais et al., 2012; Yee et al., 2013). Typically, they react to first form intermediate products with a structure akin to the initial methoxyphenol. For example, Liu et al. (2012) reported vanillin and vanillic acid as by-products of the reaction of the methoxyphenol conyferyl aldehyde in the presence of the nitrate radical. A mechanism of the formation of vanillin from larger lignin decomposition product was also described by Wong et al. (2010).

The mean ER of vanillic acid is 2.2 (1.4 – 3.8. The compound represents 0.1 % to 0.3 % of the OA concentrations at its peak concentration,during aging (Figure 8). This is within the same range of proportions observed before lights on. Similar ER are obtained for vanillin and acetovanillone (2.1 and 2.2, respectively). Syringic acid and tyrosol shows a slightly more moderate mean ER of 1.9 and the mean ER of the pyrogallol is 3.5 (2 - 5).

### 3.2.3 Secondary compounds

The identified secondary compounds include nitro-aromatic compounds such as 4-nitrocatechol (4-NC) and its methylated derivatives, 4-methyl-5-nitrocatechol (4M5NC) and 3-methylated-5-nitrocatechol (3M5NC), as well as methylglutaric acid and vanillylmandelic acid. After 4 - 6 hours of atmospheric aging, these compounds contribute 2.5 % of total OA. Nitrocatechols are the most abundant secondary compounds detected. Iinuma et al. (2010) and Kelly et al. (2010) demonstrated their formation through the oxidation of cresol in the presence of NO$_x$. The aerosol formation of varies between 7 and 65 mg kg$^{-1}$ for 4-NC, between 0.3 and 6 mg kg$^{-1}$ for 4M5NC and between 0.4 and 10 mg kg$^{-1}$ for 3M5NC. In the experiments where 4-NC is detected in the primary emission (< 0.2 % of total OA), the ER averages at 80 (15 - 260). For experiment 8 only, we observe a substantial amount of 4-NC in the primary emissions (> 1% of the POA mass concentration). The ER of the compound is 6 in this case. With regard to the alkylated nitrocatechols, five of the eleven experiments present trace levels of 4M5NC in the primary emissions. The ER averages at 17 (3 to 50). The average ER for 3M5NC is 14 (3 to 45). The concentrations of these compounds increase within the first 1 – 2 hours of atmospheric aging, and contribute as high as 5 - 6 % of the total OA mass concentration in the case of 4-NC (Figure 8), and up to 0.4 % and 0.8 % for 3M5NC and 4M5NC. The concentration of 3M5NC decreases soon after the initial peaking (by a factor of 2 to 3 near the end of the experiment), whereas the concentrations of 4-NC and 4M5NC remain stable. We hypothesize that formation and loss processes compete especially in the case of these compounds.

Nitro-aromatic compounds have been observed at sites impacted by biomass burning before (Iinuma et al., 2010, Claeys et al., 2012, Kitanovski et al., 2012, Mohr et al., 2013, Kahnt et al., 2013, Kitanovski et al., 2014, Frka et al., 2016, Gaston et al., 2016) at concentration levels typically two orders of magnitude lower than that of levoglucosan (4-NC/LG ≈



0.01 – 0.1, MNC/LG ≈ 0.01 - 0.08). Here, 4-NC/LG and MNC/LG ratios vary from 0.06 to 0.15 and 0.25 to 0.5, respectively. We hypothesize that the higher levels of 4-nitrocatechol emissions found in the chamber are the result of high concentration of $NO_x$ (Table 1) (in a range of 50 to 252 ppb, compared to 40 – 50 ppb in areas heavily impacted by biomass burning (Brulfert et al., 2005; Sandradewi et al., 2008)). Experiments conducted at higher $NO_x$ concentrations (i.e.

experiments 9 – 11 with stove C) (see Table 1, THC/$NO_x$ ≈ 5, Cresol/ $NO_x$ < 0.005) show the highest contribution of 4-NC to the OA mass concentration (3 – 6 %).

The mass spectra (EI, 70eV) of 4-nitrocatechol and the alkylated nitrocatechols are dominated by the ion $C_6H_5NO_4^+$ ($m/z$ 155) and $C_6H_2NO_4^+$ ($m/z$ 152), respectively, which are detected with the HR-ToF-AMS. Figure 10 shows the intensity of the fragments from HR-ToF-AMS measurements (expressed in µg m$^{-3}$ nitrate equivalents) against the concentrations of

the nitrocatechol compounds measured by the TAG-AMS. The best correlation is observed for 4-nitrocatechol ($R^2$ = 0.78). For alkylated nitrocatechols the correlation is slightly weaker ($R^2$ = 0.64), most probably due to the contribution of different isomers. While a proper calibration is needed, such relationships indicate that these two ions ($C_6H_5NO_4^+$ and $C_6H_2NO_4^+$) can potentially be considered as suitable tracers of nitrocatechol compounds in AMS measurements and could offer interesting perspectives for the quantification of secondary BBOA in ambient atmosphere. However, and because of possible

interferences from other aerosol sources, this assumption must be confirmed with more complex organic matrices.

Finally, this third category also includes less often reported compounds. They however represent only a small fraction of the total OA mass concentration. At its peak, the methylglutaric acid contributes less than 0.05 %. The aerosol formation varies between 0.01 and 2 mg kg$^{-1}$. For the majority of the experiments, this compound is detected only during aging. Only two experiments show traces of the compound before lights on. Their ER are 9 and 15, respectively. The

vanillylmandelic acid contributes as much as 0.2 % of the total OA. The EFs for aged emissions vary between 0.04 and 1 mg kg$^{-1}$. The compound is detected in the primary emission of all experiments but its enhancement ratio is > 10.

**4 Conclusions**

We determined the impact of burning conditions and atmospheric aging on the chemical fingerprint of the OA emitted by biomass burning by means of a TAG-AMS. We provide a detailed quantitative analysis of the evolution of the contribution

of 71 markers to the OA mass from fresh to aged emission. We draw the following conclusions:

- The relative contribution of compounds to fresh emissions can vary significantly between the different types of stoves used to generate the emission. Differences are mainly due to the quality of combustion (i.e. MCE). Smoldering combustion increases the EF of the reported compounds, but overall decreases their contribution to the total OA mass concentration. This indicates that smoldering combustion induces the emission of OA with a

30        more complex composition. The contribution of levoglucosan for instance varies from 40 % – 50 % at the highest MCE, down to 15 % at the lowest MCE.



- The resolved fraction of fresh woodstove emissions by TAG-AMS is on average 52 %, but dramatically decreases with photochemical aging to < 10% after an integrated OH exposure of $5 \times 10^6$ molecules cm$^{-3}$ h. The majority of secondary compounds thus remain unidentified.

- The contribution of a variety of tracers to the total OA mass concentration evolves substantially during aging. We establish ratios for all reported compounds as a function of the photo-oxidative processing time. For instance, the contribution by levoglucosan, the concentration of which decreases by approximately 40 % during aging, is < 10 % to the total OA mass concentration at the end of the experiment against an average contribution of 30 % for the fresh emissions.

- Nitro-aromatic compounds such as 4-nitrocatechol, 4-methyl-5-nitrocatechol (4,5-MNC) and 3-methylated-5-nitrocatechol are formed during aging. They can contribute significantly to the total OA mass concentration (2 – 4 %), and thus may serve as aged biomass burning tracers in future source apportionment studies.

These data will serve to improve our ability to properly apportion the biomass burning source in the ambient atmosphere.

**Acknowledgments**

This work was supported by the French Environment and Energy Management Agency (ADEME) project VULCAIN (grant number: 1562C0019). AB also acknowledges ADEME and the Provence-Alpes-Côte d'Azur (PACA) region for their support. CNJ and AHG acknowledge support from NSF AGS Award 1524211 and NOAA Award NA16OAR4310107. PSI acknowledges the financial contribution by the SNF project WOOSHI and the IPR-SHOP SNF starting grant.



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



**Table 1: Summary of the experiments and conditions in the chamber before lights-on. MCE stands for modified combustion efficiency and THC is total hydrocarbon.**

| Stove | Exp # | MCE | Nb of TAG-AMS samples | *[BC] ($\mu g.m^{-3}$) | *[POA] ($\mu g.m^{-3}$) | **[OA] ($\mu g.m^{-3}$) | [NOₓ] (ppb) | [THC]/[NOₓ] (ppb/ppb) | [Cresol]/[NOₓ] (ppb/ppb) |
|---|---|---|---|---|---|---|---|---|---|
| Stove A (Beech as logs) | Exp 1 | 0.87 | 6 | 16.8 | 122.3 | 495.4 | 98.0 | 31.5 | $1.1 \times 10^{-1}$ |
| | Exp 2 | 0.84 | 7 | 12.4 | 176.8 | 785.0 | 252.0 | 26.9 | $1.1 \times 10^{-1}$ |
| | Exp 3 | 0.84 | 7 | 5.6 | 71.3 | 387.9 | 90.0 | 38.5 | $1.2 \times 10^{-1}$ |
| | Exp 4 | 0.95 | 8 | 5.1 | 10.2 | 72.1 | 128.0 | 7.7 | $2.7 \times 10^{-2}$ |
| Stove B (Beech as logs) | Exp 5 | 0.83 | 7 | 4.8 | 40.7 | 143.5 | 50.0 | 47.2 | $1.2 \times 10^{-1}$ |
| | Exp 6 | 0.89 | 7 | 13.0 | 37.7 | 202.1 | 119.0 | 18.1 | $5.0 \times 10^{-2}$ |
| | Exp 7 | 0.84 | 6 | 5.8 | 44.6 | 289.1 | 114.0 | 24.3 | $7.2 \times 10^{-2}$ |
| | Exp 8 | 0.93 | 7 | 4.4 | 9.3 | 53.1 | 80.0 | 19.6 | $4.5 \times 10^{-2}$ |
| Stove C (Softwood pellets) | Exp 9 | > 0.99 | 5 | 107.4 | 10.0 | 18.7 | 161.0 | 5.2 | $1.2 \times 10^{-3}$ |
| | Exp 10 | > 0.99 | 6 | 130.1 | 10.5 | 19.3 | 205.0 | 5.8 | $4.7 \times 10^{-4}$ |
| | Exp 11 | > 0.99 | 5 | 144.3 | 10.2 | 21.7 | 228.0 | 5.9 | $3.5 \times 10^{-4}$ |

*values retrieved just before lights on

**values retrieved at integrated OH exposure $= 5 \times 10^6$ molecules cm$^{-3}$ hour



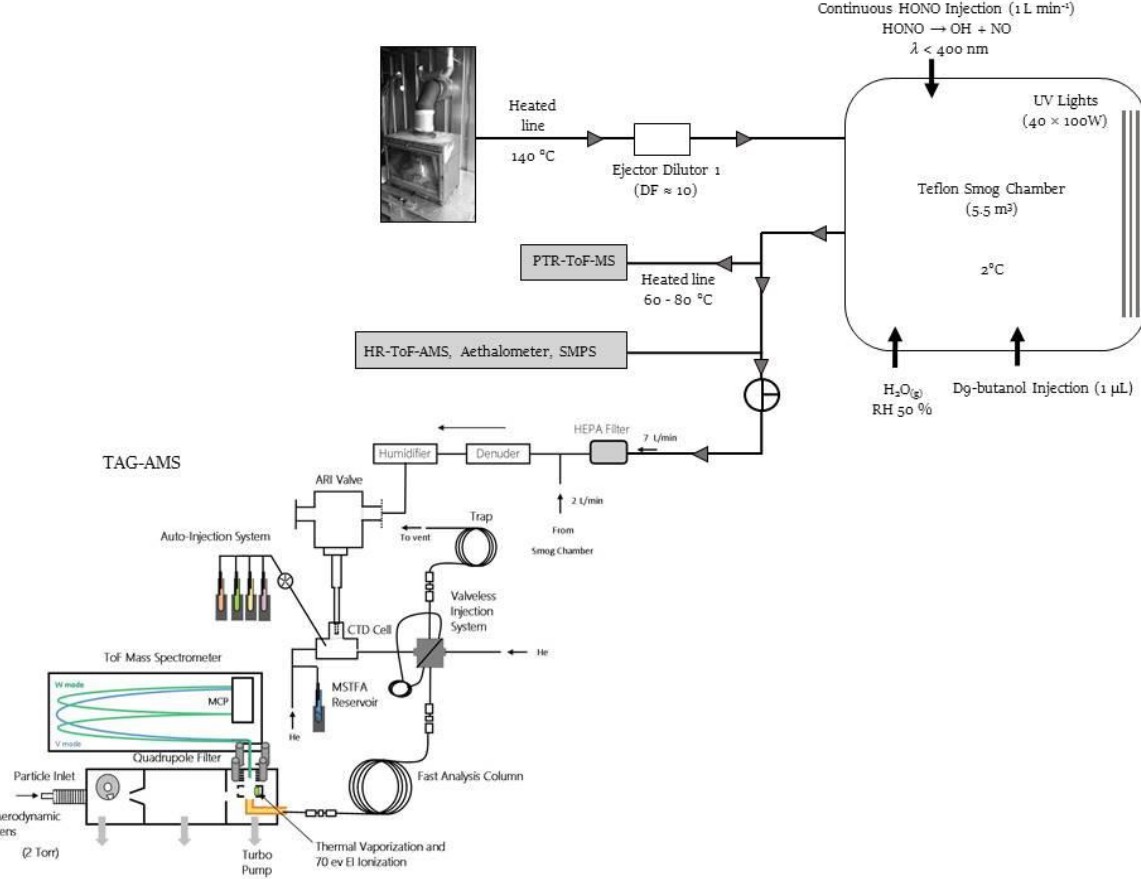

**Figure 1: Simplified scheme of the set-up with the TAG-AMS coupled to the chamber.**





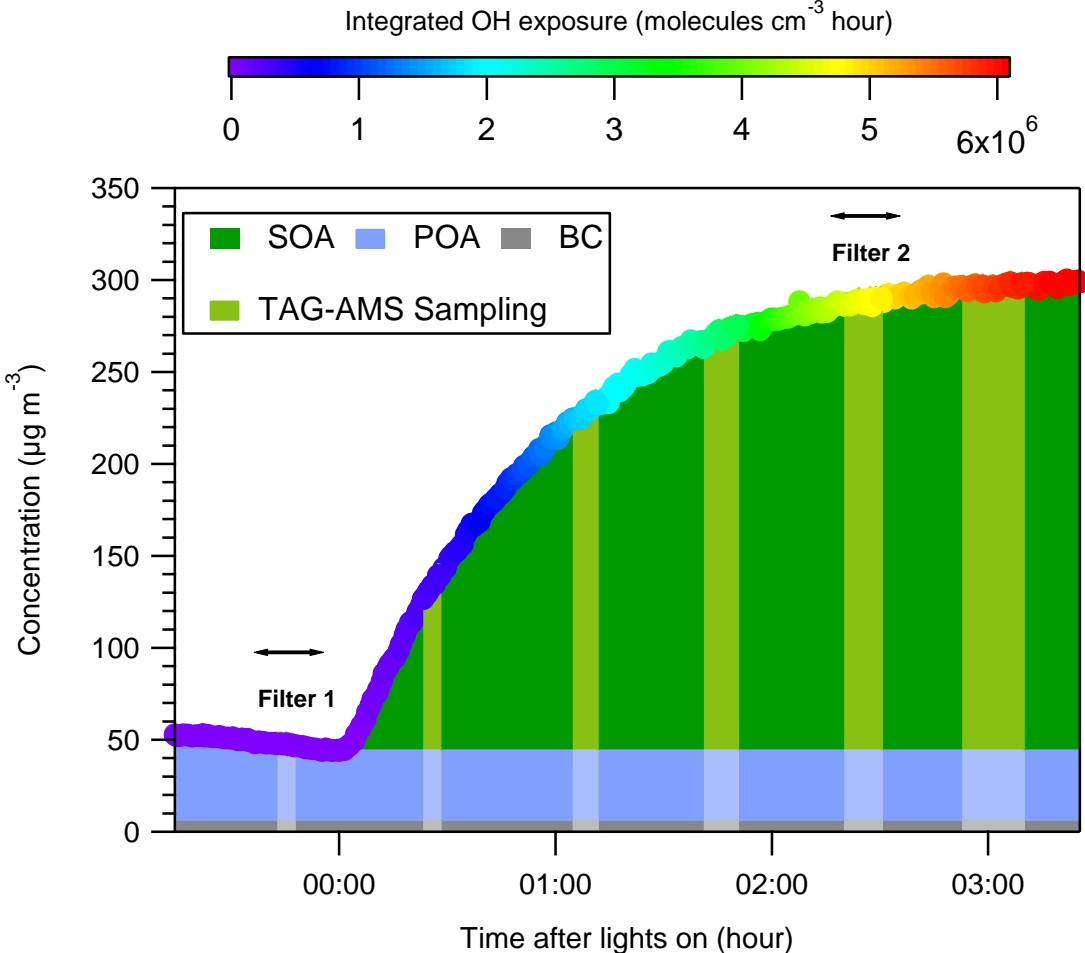

**Figure 2: Example of the TAG-AMS and offline (quartz fiber filters) measurements sampling schedule during an aging experiment conducted at the PSI atmospheric chamber with emissions generated from woodstove appliances (Exp. 7). One sampling with the TAG-AMS is carried out before lights on to characterize POA. At $t = 0$ h, HONO is injected and the lights turned on to initiate photo-chemistry. 4 to 6 samples were taken during the aging period.**



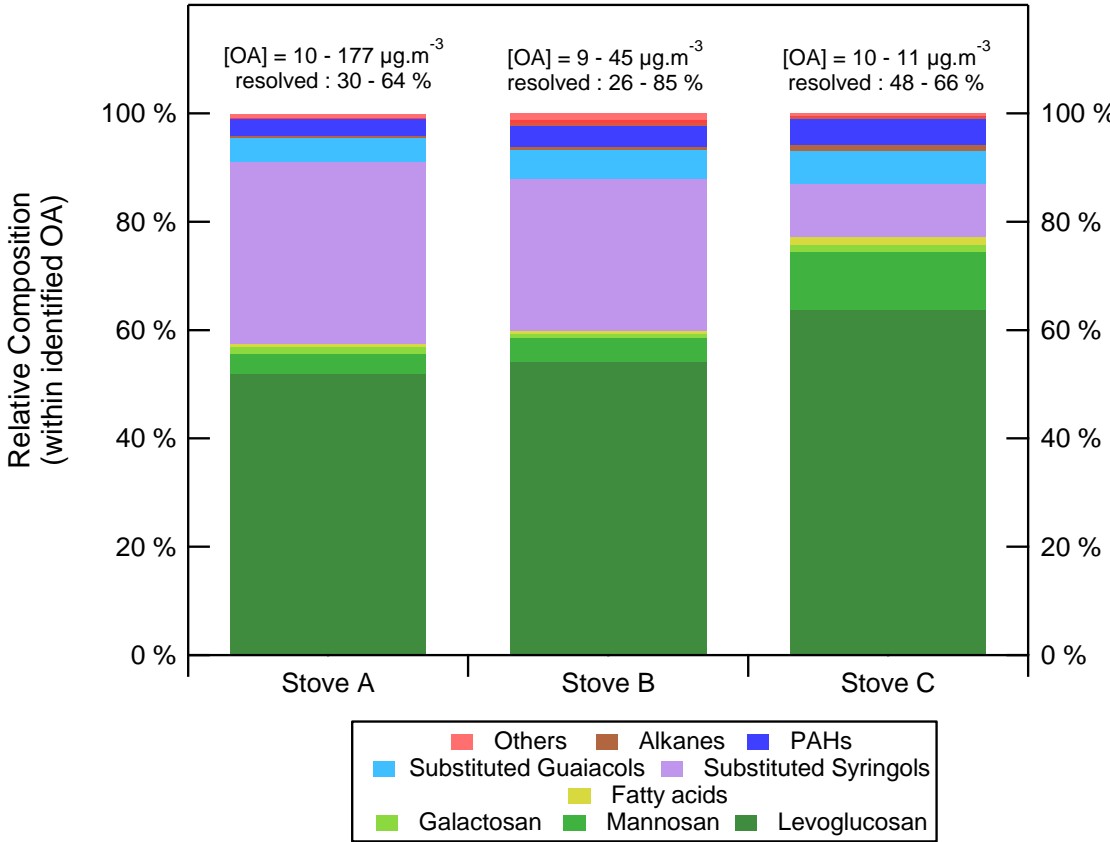

**Figure 3: Averaged relative composition of the resolved OA mass fraction by TAG-AMS for the three different appliances used to generate the primary emissions.**



**Figure 4: Emissions factors of POA, levoglucosan, and methoxyphenols, as well as their contribution to the total OA as a function of the modified combustion efficiency (MCE). Details on how the MCE was calculated are given in Bertrand et al. (2017).**

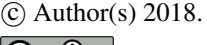



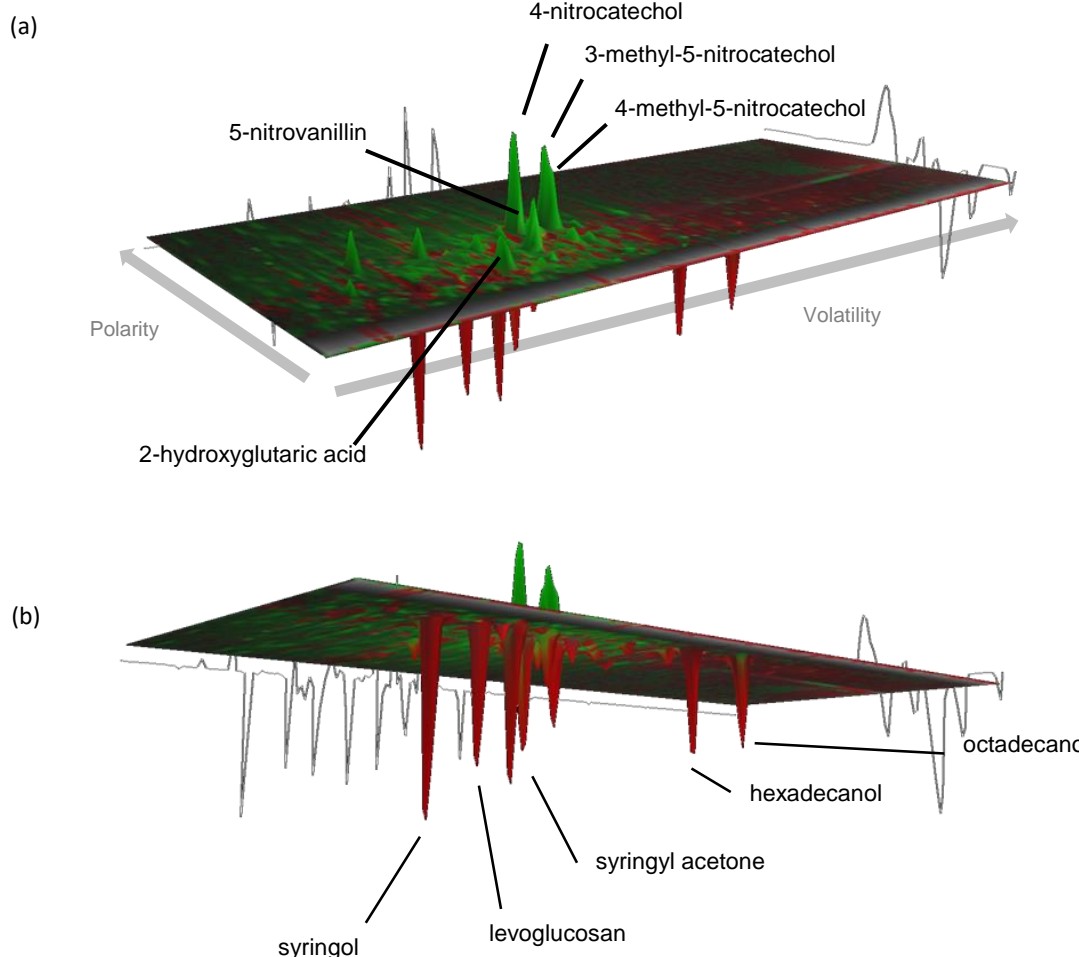

**Figure 5: Three-dimensional representation of the differences between aged (integrated OH exposure of $5 \times 10^6$ molecules cm$^{-3}$ h) and primary OA 2D-GC chromatograms normalized to the internal standard responses (See SI for additional details regarding 2D-GC analysis). Highlighted in green are the all net-positive compounds (signal increasing during aging), and in red are all the net-negative compounds (signal decreasing during aging) (Experiment 5).**





**Figure 6: Evolution of the chemical fingerprint of the organic aerosol emitted by biomass burning during atmospheric aging (Experiment 2).**





**Figure 7: Mean enhancement ratios (ER) of individual compounds in biomass burning emissions. The ER is defined as the pWLC concentration measured in the particle phase within a time range corresponding to an integrated OH exposure of $4 \times 10^6$ - $6 \times 10^6$ molecules cm$^{-3}$ hour (bin 4, Figure 6) and divided by the concentration in the particle phase measured at $t_0$ or before lights on (i.e. ER = 0.8 means a loss of 20 % after pWLC). Note that for the secondary compounds (colored in red) observed after the start of the photo-oxidative process only, the ER is calculated using the detection limit.**



**Table 2: pWLC contribution of the compounds to the total OA mass concentration at different times of the photo-oxidative process. This table only contains a short list of compounds and average values. A full list, with minimum and maximum values is also available in the SI.**

| Integrated OH exposure (molecules cm⁻³ hour) (n = nb samples) | Bin 0 0 n = 11 | Bin 1 > 0 - 5 x 10⁵ n = 8 | Bin 2 0.5 - 2 x 10⁶ n = 12 | Bin 3 2 - 4 x 10⁶ n = 15 | Bin 4 4 - 6 x 10⁶ n = 16 | Bin 5 6 - 7.5 x 10⁶ n = 7 | Bin 6 7.5 - 9 x 10⁶ n = 2 |
|---|---|---|---|---|---|---|---|
| Total OA (µg.m⁻³) (min - max) | 9 - 177 | 32 - 409 | 13 - 715 | 18 - 790 | 19 - 774 | 18 - 504 | 50 - 517 |
| **Total Contribution* (% OA)** | **52** | **20** | **20** | **14** | **10** | **13** | **9** |
| *Primary Compounds (%OA)* | | | | | | | |
| Levoglucosan | 29.2 | 9.5 | 9.6 | 6.3 | 4.6 | 6.0 | 3.5 |
| Mannosan | 3.2 | 0.8 | 1.4 | 0.9 | 0.5 | 0.8 | 0.3 |
| Fluorene | 0.15 | 0.06 | 0.07 | 0.05 | 0.03 | 0.05 | 0.08 |
| Phenanthrene | 0.41 | 0.18 | 0.19 | 0.16 | 0.12 | 0.16 | 0.1 |
| Anthracene | 0.08 | 0.04 | 0.03 | 0.02 | 0.02 | 0.02 | 0.02 |
| Fluoranthrene | 0.22 | 0.07 | 0.14 | 0.13 | 0.1 | 0.2 | 0.14 |
| Pyrene | 0.08 | 0.02 | 0.05 | 0.05 | 0.04 | 0.07 | 0.05 |
| 1,2-Acenaphthylenone | 0.31 | 0.14 | 0.15 | 0.10 | 0.08 | 0.11 | 0.08 |
| 9H-Fluoren-9-one | 0.03 | 0.01 | 0.02 | 0.02 | 0.01 | 0.02 | 0.02 |
| Cyclopenta[d,e,f]phenanthrene | 0.03 | < 0.01 | 0.01 | 0.01 | < 0.01 | 0.01 | 0.01 |
| 3-Guaiacylpropanol | 0.28 | 0.10 | 0.05 | 0.03 | 0.03 | 0.02 | 0.03 |
| Conyferyl Aldehyde | 0.29 | 0.07 | 0.04 | 0.03 | 0.02 | 0.02 | 0.02 |
| Syringaldehyde | 2.7 | 1.7 | 1.1 | 0.7 | 0.5 | 0.6 | 0.6 |
| Syringol | 1.43 | 1.08 | 0.59 | 0.25 | 0.16 | 0.19 | 0.13 |
| Methylsyringol | 0.09 | 0.06 | 0.03 | 0.01 | 0.01 | 0.01 | 0.01 |
| Acetosyringone | 0.41 | 0.20 | 0.11 | 0.06 | 0.05 | 0.05 | 0.05 |
| Isoeugenol | 1.22 | 0.67 | 0.27 | 0.09 | 0.08 | 0.09 | 0.09 |
| Syringyl Acetone | 4.6 | 1.5 | 0.4 | 0.2 | 0.2 | 0.2 | 0.3 |
| Propionyl Syringol | 0.7 | 0.3 | 0.2 | 0.2 | 0.1 | 0.1 | 0.1 |
| Synapyl Aldehyde | 0.9 | 0.2 | 0.09 | 0.05 | 0.03 | 0.04 | 0.04 |
| Palmitic Acid | 0.28 | 0.09 | 0.15 | 0.09 | 0.06 | 0.11 | 0.04 |
| Stearic Acid | 0.11 | 0.05 | 0.06 | 0.04 | 0.03 | 0.05 | 0.02 |
| *Non-Conventional Primary Compounds (% OA)* | | | | | | | |
| Vanillin | 1.7 | 0.8 | 1.1 | 0.9 | 0.6 | 1.0 | 0.7 |
| Acetovanillone | 0.29 | 0.13 | 0.17 | 0.14 | 0.14 | 0.14 | 0.1 |
| Vanillic Acid | 0.24 | 0.12 | 0.16 | 0.14 | 0.1 | 0.14 | 0.09 |
| Syringic Acid | 0.17 | 0.09 | 0.11 | 0.08 | 0.06 | 0.08 | 0.05 |
| Pyrogallol | 0.02 | 0.02 | 0.03 | 0.02 | 0.02 | 0.02 | 0.01 |
| Tyrosol | 0.36 | 0.16 | 0.17 | 0.12 | 0.09 | 0.12 | 0.15 |
| *Secondary Compounds (%OA)* | | | | | | | |
| 4 Nitrocatechol | 0.2 | 0.8 | 1.7 | 2.4 | 1.8 | 1.9 | 1.1 |
| 4-Methyl-5-Nitrocatechol | 0.05 | 0.15 | 0.2 | 0.18 | 0.14 | 0.15 | 0.11 |
| 3-Methyl-5-Nitrocatechol | 0.15 | 0.48 | 0.45 | 0.32 | 0.25 | 0.24 | 0.19 |
| Vanillylmandelic acid | 0.02 | 0.08 | 0.06 | 0.04 | 0.03 | 0.03 | 0.02 |
| Methylglutaric acid | < 0.01 | 0.02 | 0.02 | 0.02 | 0.02 | 0.02 | 0.02 |

*Sum from the contribution of all compounds averaged within the bins





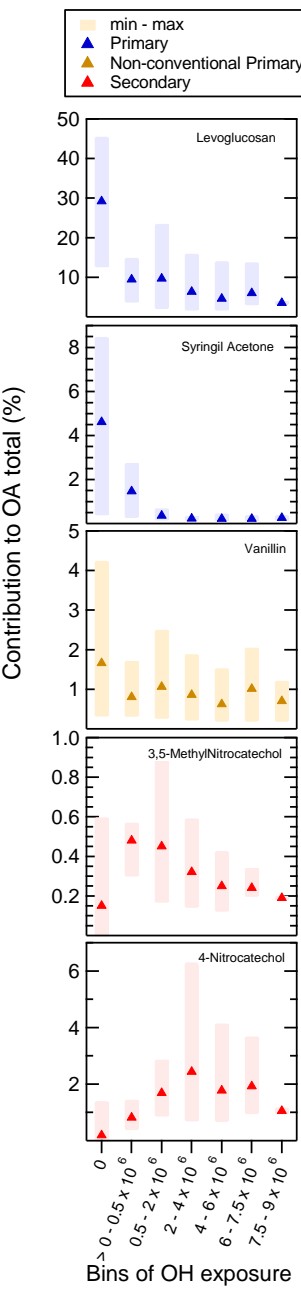

**Figure 8: Evolution of the contribution of several markers to the OA$_{(t)}$ mass concentration at different time points in the photo-oxidation process. The marker represents the mean value over all experiments within a specific range of integrated OH exposure. The box includes the minimum and maximum within the same range.**





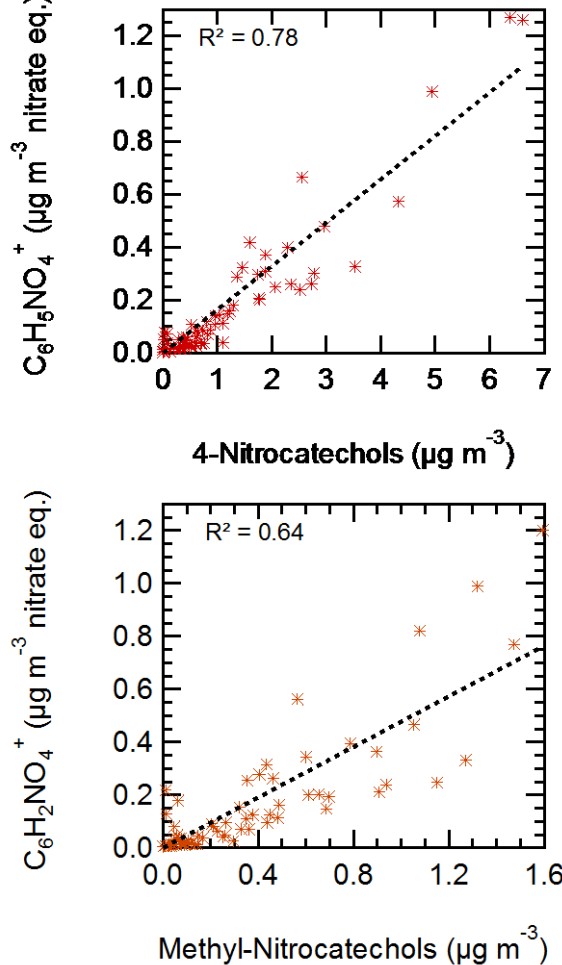

**Figure 9: HR-ToF-AMS measurements against TAG-AMS measurements. $C_6H_5NO_4^+$ (*m/z* 155.022) vs. 4-nitrocatechol (top), and $C_6H_2NO_4^+$ (*m/z* 151.098) vs. alkylated nitrocatechols (3-methyl-5-nitrocatechol and 4-methyl-5-nitrocatechol) (bottom), during the**
5  **aging of biomass burning emissions.**