# Peer review of "Evolution of the chemical fingerprint of biomass burning organic aerosol during aging"

_Atmospheric Chemistry and Physics, 2017_

## Referee Comment (RC1) · Anonymous Referee #1 · 2 Feb 2018

General Comments

This study investigated the chemical composition of biomass burning organic aerosol using online analysis techniques, principally Thermal Desorption Aerosol Gas Chromatography coupled to an Aerosol Mass Spectrometer (TAG-AMS). Solid wood fuels were burned in stoves and emissions were characterized. The smoke was exposed to hydroxyl radicals (OH), which increased organic aerosol (OA) mass and also reacted with components of the primary smoke aerosol. A variety of compounds were characterized and quantified, and the manuscript documents the quantities thoroughly, if not in such a dense manner as to make the paper challenging to read. Two main scientific comments arose during this review:

(1) The unidentified fraction of OA should be treated more carefully in the presentation

of results. It is critical for the community to understand what was, and what was not, measured. In some cases, the treatment of unidentified fraction of OA in the text obscures the meaning of the results, which has been outlined in at least one comment below.

(2) The authors should be careful in their treatment of Stoves A and B versus Stove C. The authors describe their resolution of different burning conditions, however different fuels were used. At another point in the manuscript, a difference between Stove A/B and Stove C is attributed to the difference in fuel. Since both burning conditions and fuel type were different between the 'smoldering' and 'flaming' experiments, the authors should be vigilant and conservative in their interpretation.

With these concerns described and detailed below, it is the judgement of this reviewer that the manuscript is publishable in Atmospheric Chemistry and Physics with relatively minor revisions.

Specific Comments

Page 3, Lines 18-30: This introductory discussion would be improved by re-organizing the topics. It is suggested that after introducing levoglucosan, the possibility that it can be oxidatively degraded in particles should be discussed, followed by a discussion of the secondary OA mass production from biomass burning emissions. The study investigates the topics of (1) emission factors for POA, (2) oxidative degradation of tracers, and (3) changes in OA composition due to secondary oxidation. The introduction should also follow this logical progression.

Page 3, Line 21-23: Is this statement meant specifically for biomass burning emissions? Please clarify.

Figure 2: It appears that TAG-AMS sampling is represented by both light green and light blue. If this is the case, please edit the legend to clarify.

Page 4, Section 2.1: How was the total OA quantified? If by HR-ToF-AMS, then this

should be clearly specified and qualified by any sampling biases.

Page 6, Lines 17-21: This can be removed from the methods section as it is actually a statement of the results (and will, indeed, be discussed at great length in the results section).

Equation 1 (and supporting text): What value of k_wall/p was used?

Page 7, Lines 28-29: Please clarify what is meant by "we identify between 26-85% of the POA mass concentration". Ensure that the manuscript clearly identifies how total OA was quantified. It is assumed that the 'identified OA' is the mass concentration quantified by TAG-AMS.

Figure 3: Showing this figure with an 'unidentified' section of the bar would be more appropriate, since different amounts of OA were identifiable in each experiment.

Page 8, Lines 2-3: The text states "The MCE values for the experiments conducted with this stove [the pellet stove?] are 0.97," but Table 1 shows "MCE > 0.99" for pellet stove experiments. Which of these values is correct?

Page 8, Lines 14-17: Please clarify (with evidence and explanation) that the nature of the artifacts that should have been associated with the prior studies based on methodology are actually consistent with the differences between the prior studies and the present study, as is vaguely suggested in the submitted manuscript.

Page 8, Line 27: Methoxyphenols are not the 'predominant' class of species in the primary BBOA – Figure 3 clearly shows that anhydrosugars comprise the predominant fraction of the identifiable OA.

Page 9, Lines 5-8: This sentence embodies an important criticism of the data interpretation within this study. The authors describe differences between the softwood and hardwood samples. At the same time, hardwoods and softwood fuels were burned under different conditions. The authors seem to interpret differences between the log wood stoves and the pellet stove as differences in fuel at this point, while at other

points, the differences are attributed to burning conditions. If the interpretation given here is thought to be accurate, then an explanation for why burning conditions don't influence the observed difference must be described.

Section 3.2.1: The authors state that the primary compounds initially represent 48% of the POA, but after oxidation (4-6 hours) they represent < 8%... of what? Of POA? Of total OA? Can this just be described by the addition of SOA mass to the particles? Figure 6 shows that the relative contribution of levoglucosan is somewhat stable around 40% of identified OA throughout the experiment. Please clarify what is being quantified and to what fraction of the aerosol the percentage is referring. It seems from this text that the mass concentration of levoglucosan, for example, decreased substantially during exposure to OH, but Figure 6 suggests that it is relatively stable against aging based on how the data are shown. Then again, Figure 7 shows the ER = 0.6, so the actual concentration is decreasing during aging. Figure 8 shows that levoglucosan is decreasing as a fraction of total OA. Basically, Figure 6 seems inconsistent with Figs 7 and 8. Is the change in the identifiable fraction playing a role? It is very likely that the data are somehow telling a consistent story, but it is not presented clearly. [Note that this comment explicitly references levoglucosan, but the comment also applies to other Primary Compounds.]

Page 12, Line 33 – Page 13, Line 2: What is the relevance of the ratios of nitrocatechols to levoglucosan, since it is known that the latter is consumed during OH exposure?

Page 13, Line 3: Higher levels of 4-NC are discussed with reference to prior observations, however far larger discrepancies between the prior observations and the present study exist for MNC/LG. The 4-NC/LG ratio range at least overlaps the prior studies, but the MNC/LG ratios are shifted by about a factor of 10, yet are not discussed.

Page 13, Line 7: Does "the mass spectra (EI, 70 eV)" refer to those obtained from a database? Please clarify. If from a database, please cite the reference.

Page 13, Line 23-24: The impact of burning conditions is less clear since the fuel was

different for Stove C. Please clarify this issue in all instances within the manuscript, including Page 9, Lines 5-8 as discussed in a comment above.

Page 14, Lines 1-3: How might measurements be improved to detect and quantify a greater fraction of the POA and SOA mass? Can this be inferred from the limitations of TAG-AMS? [A discussion of this topic may not be entirely appropriate for the conclusion section, but may be a worthwhile topic to include elsewhere in the manuscript.]

Technical corrections –

Page 3, Line 4: correct the in-line citation

Page 9, Line 3: correct the in-line citation

Page 9, Line 21: consider beginning a new paragraph for fatty acids

Page 13, Line 8: "Figure 10" should be "Figure 9"

––––––––––––––––––––––––––––––––––

---

## Referee Comment (RC2) · Anonymous Referee #2 · 12 Feb 2018

Review.

Evolution of the chemical fingerprint of biomass burning organic aerosol during aging

Overall.

This paper uses TAG-AMS to quantify compounds in organic aerosol from two log stoves and one pellet stove after emission and after equivalent atmospheric aging in a chamber. Three categories of compounds are described, and the emissions factors and emission profiles of biomass burning tracers are measured. The authors quantify how the OA changes after atmospheric aging. This paper is generally well-written but would benefit from a read through for typos (see examples in editorial section). Additionally, more discussion on implications of this study could be included in the

conclusions. Some suggestions for improvement are below, but overall the paper is interesting and worthy of publication.

General.

The variability in the total OA and the resolved OA for each stove type is interesting and should be further explained. It is unclear that the three or four samples from each should be simply averaged as was done in Figure 3. That resulted in one composition for each stove type without anything showing the potential variability in the composition for each stove. The discussion in the text regarding this composition should clarify if it is in regard to the total POA or the resolved fraction of POA. This should be stated clearly and be used consistently. Specific comments are below.

In the section on atmospheric aging, the authors should be more explicit about what measurements went into the results of the gaining experiment. It seems as though all of the OA was aged, but there is no explicit mention of which stove was used to generate the POA that was then aged. The initial composition of the POA could influence the SOA composition, so there should be more discussion of this, unless the resolved fraction is too small to draw conclusions.

Specific.

P. 3, L. 6-9: It would be better to incorporate these references into the paragraph and the following sentences. More detail is included later about the biomass burning, but not the other sources.

P. 3, L. 17: This is redundant with the previous mention of Levoglucosan at L. 11. Remove or rephrase.

P. 4 – Is particle bounce a factor in the TAG-AMS? Was that taken into account in the calculations of EFs? How does the TAG-AMS treat organics on black carbon? Can the black carbon be detected?

P. 4, L. 30: Comment on what is causing such a large range in OA concentration.

Does this have anything to do with the amount of time it was burning? How was this controlled for? Does the emission factor of each stove change with the amount of time the fuel has been burning? How did the emissions change over the 30 minutes they were stabilizing in the chamber? More details should be added here.

Table 1: What is the difference between "in the chamber before lights-on" and "values retrieved just before lights on"? What is the timing difference between the [BC] and the [NOX]? Why are the concentrations of BC, etc. not consistent? What other conditions were changed?

Figure 1: The words are all very small. It might be possible to shift so that the figure takes up less space (i.e. is less diagonal).

P. 5, L. 16: "discrimination of all fragments below m/z 10" – Explain or rephrase.

P. 6, L. 10: Wall losses decreased the concentration of aerosol, and this was accounted for qualitatively by increasing the sampling length as time progressed. Mention here that the quantitative estimates for wall losses are included in Section 2.4. These questions came up here before reaching Section 2.4: Are there any calculations or quantitative estimates of the wall losses? How much time was added to each sampling period, and how was that calculated?

Figure 2: The legend is somewhat confusing. Suggest removing "TAG-AMS Sampling" and changing the caption to be "Figure 2: Example of the TAG-AMS (light shading) and . . ."

P. 6, L. 11: Was this an automated matching process? What confirmed a match? Comment on how were the initial standards and NIST spectra were selected for comparison.

P. 6, L. 30: What is eBC? Write out or explain the first time it is used. On the next page, it is listed as black carbon. In equation 2, it is written as just "BC". Make sure these are consistent, and if they are different, explain what that difference is.

Equation 1: Remove the "." – not standard formatting for equations.

P. 6 – Is there any preferential wall loss based on composition? The only aerosol tested is eBC. Is that representative of all of the aerosol?

P. 7, L. 7: How reasonable is that assumption? Was it tested?

P. 7, L. 19: Clarify or rephrase the discussion of the symbols in Equation 2. "Here refers to the concentrations" – but really, it refers to the change in the mass of each.

Figure 3: Reorganize the legend so that they are left justified. (If this is based on the groups discussed in the text, then include headings for each type). This figure would also be more useful if it could also demonstrate the fraction of total OA that it represents. It would also be nice to have some error bars or some representation of uncertainty. The fact that for Stove B, 26 to 85% of the OA was resolved at any time could also mean that there is large variability in the relative fractions. It appears that the main point of the graph is that Stove A and Stove B have similar compositions. But it would be good to see some representation of the differences in their total OA concentrations and the amount of OA resolved for each. In addition to or instead of these average bar plots, I suggest including a figure with a bar for each sample, grouped by stove type, with the y-axis representing the fraction of total OA resolved, instead.

P. 7, L. 28: Explain where the 26-85% of the number for the 26-85% of the mass comes from. In Figure 3, are the ranges due to averages taken for each sample? That is a lot of variability. That variability is in itself interesting and should be discussed.

P. 8, L. 4: This statement is not consistent with the figure. There is huge variability in the fractions of OA identified in all three stoves. Stove B has the lowest low range but also highest high range, so it is an overgeneralization to say that Stoves A and B are less resolved. Or, more data and discussion needs to be added to support that.

Figure 4: Remove the gap between a/b and c/d and the x-axis labels, since they are the

same as those at the bottom for e/f. Why are R values on c/d and not on the others? Remove them or add the others to be consistent.

P. 8 – Are these fractions of total POA or resolved POA? There is a huge range in total POA as well as resolved POA, and there appears to be a range in these sugars as well. If the standard deviations are included in the comparison, there is no statistical difference in the percentages of anhydrosugars from the three stoves.

P. 9, L. 27: It is unclear how Figure 5 illustrates that.

Figure 5: State the difference between a and b in the caption. This figure could also be moved to the supplement. It is only referenced once.

Table 2: Remove "Bin" labels, or explain them. It seems like enough for each column to have integrated OH exposure. Is this for one sample of one of the stoves or an average? It is unclear what experimental set up this data came from.

P. 10 – Clarify the units of OH exposure. This may take a new reader some time to interpret. The "integrated OH exposure" seems to be just the OH concentration (molecules/cm3) multiplied by the time exposed (hours).

Figure 6: Same legend comment as Figure 3. What POA was used in this figure? What stove is this from?

P 10 – Why does the fraction of identified OA decrease with OH exposure? This almost implies an inverse correlation between OA mass and fraction of identified OA. It seems like the OA produced at the very end is then all unidentifiable.

Figure 7: This figure is interesting, and the split between the red and blue is nice. But having the primary compounds with upside down bars make them seem negative. Additionally, the ER ratio increases from left to right, but the bars decrease in size, which is counterintuitive. The caption should state what the error bars are calculated from (and the text could comment on why they vary significantly).

P. 11, L. 19: Interesting discussion! Figure 9: In the text, this is listed as Figure 10. What is the explanation for the difference between the two measurement types? It would be good to show the slope here, in addition to the R squared value.

P. 14 – Add a couple more sentences about the implications of these results and why it is important that they were measured.

Editorial: P. 2, L. 3: Add "of" after "evolution" P. 2, L. 6: Remove redundant "emission" P. 3, L. 21: Change "during" to "due to" P. 8, L. 19: Figure 6? Should this be Figure 4? P. 10, L. 23: Add "with" after "along" P. 13, L. 8: Change Figure 10 to Figure 9

―――――――――――――――――

---

## Author Comment (AC1) · 26 Apr 2018

We thank the Referee for the careful revision and comments which helped to improve the overall quality of the manuscript. A point-by-point answer (in regular typeset) to the referee's remarks (in the *italic typeset*) follows, while changes to the manuscript are indicated in blue font. In the following document, lines references refer to the manuscript version reviewed by the anonymous referee.

*Anonymous Referee #1

*General Comments This study investigated the chemical composition of biomass burning organic aerosol using online analysis techniques, principally Thermal Desorption Aerosol Gas Chromatography coupled to an Aerosol Mass Spectrometer (TAG-AMS). Solid wood fuels were burned in stoves and emissions were characterized. The smoke was exposed to hydroxyl radicals (OH), which increased organic aerosol (OA) mass and also reacted with components of the primary smoke aerosol. A variety of compounds were characterized and quantified, and the manuscript documents the quantities thoroughly, if not in such a dense manner as to make the paper challenging to read. Two main scientific comments arose during this review:*

*(1) The unidentified fraction of OA should be treated more carefully in the presentation of results. It is critical for the community to understand what was, and what was not, measured. In some cases, the treatment of unidentified fraction of OA in the text obscures the meaning of the results, which has been outlined in at least one comment below.*

We thank anonymous reviews #1 for his comments. We modified the text page 4 line 10 – 15 to define the following: total OA (OA quantified by HR-ToF-AMS) and identified OA fraction (fraction of total OA identified and quantified by TAG-AMS). In addition, the manuscript was revised in order to explicitly state any time it was needed whether we talked about the contribution of a marker to the total POA mass concentration, to the total OA mass concentration, or to the identified OA fraction.

*(2) The authors should be careful in their treatment of Stoves A and B versus Stove C. The authors describe their resolution of different burning conditions, however different fuels were used. At another point in the manuscript, a difference between Stove A/B and Stove C is attributed to the difference in fuel. Since both burning conditions and fuel type were different between the 'smoldering' and 'flaming' experiments, the authors should be vigilant and conservative in their interpretation. With these concerns described and detailed below, it is the judgement of this reviewer that the manuscript is publishable in Atmospheric Chemistry and Physics with relatively minor revisions.*

We agree with reviewer #1 that our discussion can be more conservative all throughout this section of the results regarding the influence of the fuels. As the influence of fuels is already extensively described in the literature, we decided to put more emphasis on the impact of burning conditions on both emission factors (EF) and relative composition of OA. From our set of experiments, the burning conditions appear as the main parameter driving both EF and relative contributions for most of the organic markers quantified here. Furthermore, as the stoves represent different technology (logwood stoves for hardwood, and pellet burner for softwood), a specific and relevant discussion on the influence of the fuel burnt is complex if not impossible.

Overall, as evidenced by Figure 4, EFs increase as the MCE decreases whatever the nature of the fuel burnt with the notable exception of methoxyphenols for which the nature of the fuel is

sensitive. But while EF increase, we observe that lower MCE favors lower relative contribution of the compounds quantified here which implies the emissions of compounds not identified nor quantified in our experiments (most probably higher MW compounds). This result is, from our point of view, of prime importance for source apportionment studies.

Therefore we revised the manuscript in the following manner:

Line 27, page 7:

[revised manuscript text omitted]

*Specific Comments*

*Page 3, Lines 18-30: This introductory discussion would be improved by re-organizing the topics. It is suggested that after introducing levoglucosan, the possibility that it can be oxidatively degraded in particles should be discussed, followed by a discussion of the secondary OA mass production from biomass burning emissions. The study investigates the topics of (1) emission factors for POA, (2) oxidative degradation of tracers, and (3) changes in OA composition due to secondary oxidation. The introduction should also follow this logical progression.*

The introduction was revised entirely in the following manner:

[revised manuscript text omitted]

*Page 3, Line 21-23: Is this statement meant specifically for biomass burning emissions? Please clarify.*

Anonymous reviewer #1 is correct, this statement has been corrected.

*Figure 2: It appears that TAG-AMS sampling is represented by both light green and light blue. If this is the case, please edit the le gend to clarify.*

Figure 2 in the manuscript was edited to clarify the TAG-AMS sampling period (Figure 1)

[Figure]

**Figure 2: Example of the TAG-AMS and offline (quartz fiber filters) measurements sampling schedule during an aging experiment conducted at the PSI atmospheric chamber with emissions generated from woodstove appliances (Exp. 7, stove B).**

*Page 4, Section 2.1: How was the total OA quantified? If by HR-ToF-AMS, then this should be clearly specified and qualified by any sampling biases.*

OA was indeed quantified using the HR-ToF-AMS. Line 15, page 4 was modified to include this information:

A HR-ToF-AMS (Aerodyne Research Inc.) [was used] for the bulk-condensed chemical composition and quantification of the non-refractory fraction of the aerosol (OA, sulfate, nitrate, ammonium).

*Page 6, Lines 17-21: This can be removed from the methods section as it is actually a statement of the results (and will, indeed, be discussed at great length in the results section).*

Following the suggestion by anonymous reviewer #1, this section was now moved in introduction of the results section.

*Equation 1 (and supporting text): What value of k_wall/p was used?*

The particle half time was comprised between 2.1 and 3.5 h or a kwall/p of (0.003 – 0.006 min$^{-1}$). We included this information line 4 page 7:

t is the time since lights on (in min) and $kwall/p$ is the eBC wall loss rate constant (0.003 – 0.006 in min$^{-1}$)

*Page 7, Lines 28-29: Please clarify what is meant by "we identify between 26-85% of the POA mass concentration". Ensure that the manuscript clearly identifies how total OA was quantified. It is assumed that the 'identified OA' is the mass concentration quantified by TAG-AMS.*

The total OA mass concentration was quantified with the HR-ToF-AMS (9 – 177 ug m-3 for primary emissions), out of which the TAG-AMS is capable of identifying between 26 – 85 % (for primary emissions). Identified OA corresponds to the mass concentration quantified by TAG-AMS.

This statement was moved line 2, page 5 of the revised manuscript. For clarification purposes, and in addition to our changes already made in line 15, page 4 (as mentioned above), we modified the statement as follow.

These compounds contribute together between 26 - 85 % of the total POA mass concentration measured by HR-ToF-AMS (Figure 3).

*Figure 3: Showing this figure with an 'unidentified' section of the bar would be more appropriate, since different amounts of OA were identifiable in each experiment.*

Following in the comments by both reviewer #1 and 2 we revised the figure (Figure 3) and show now the contribution of the compounds to the total POA mass concentration measured by HR-ToF-AMS for all the experiments.

[Figure]

**Figure 3: Contribution of the organic markers measured by TAG-AMS to the total POA mass concentration (indicated on graph) measured by HR-ToF-AMS, for all experiments.**

*Page 8, Lines 2-3: The text states "The MCE values for the experiments conducted with this stove [the pellet stove?] are 0.97," but Table 1 shows "MCE > 0.99" for pellet stove experiments. Which of these values is correct?*

The MCE as indicated in the text and on the plot (Figure 4) are the correct values. We apologize for this mistake. Please find the correct version of the Table below:

**Table 1: Summary of the experiments and conditions in the chamber before lights-on. MCE stands for modified combustion efficiency and THC is total hydrocarbon.**

| Stove | Exp # | MCE | Nb of TAG-AMS samples | *[eBC] ($\mu g.m^{-3}$) | *[POA] ($\mu g.m^{-3}$) | **[OA] ($\mu g.m^{-3}$) | *[$NO_x$] (ppb) | [THC]/[$NO_x$] (ppb/ppb) | [Cresol]/[$NO_x$] (ppb/ppb) |
|---|---|---|---|---|---|---|---|---|---|
| Stove A (Beech as logs) | Exp 1 | 0.85 | 6 | 17 | 122 | 495 | 98 | 31.5 | $1.1 \times 10^{-1}$ |
| | Exp 2 | 0.84 | 7 | 12 | 177 | 785 | 252 | 26.9 | $1.1 \times 10^{-1}$ |
| | Exp 3 | 0.83 | 7 | 6 | 71 | 388 | 90 | 38.5 | $1.2 \times 10^{-1}$ |
| | Exp 4 | 0.91 | 8 | 5 | 10 | 72 | 128 | 7.7 | $2.7 \times 10^{-2}$ |
| Stove B (Beech as logs) | Exp 5 | 0.80 | 7 | 5 | 41 | 143 | 50 | 47.2 | $1.2 \times 10^{-1}$ |
| | Exp 6 | 0.87 | 7 | 13 | 38 | 202 | 119 | 18.1 | $5.0 \times 10^{-2}$ |
| | Exp 7 | 0.82 | 6 | 6 | 45 | 289 | 114 | 24.3 | $7.2 \times 10^{-2}$ |
| | Exp 8 | 0.90 | 7 | 4 | 9 | 53 | 80 | 19.6 | $4.5 \times 10^{-2}$ |
| Stove C (Softwood pellets) | Exp 9 | 0.97 | 5 | 107 | 10 | 19 | 161 | 5.2 | $1.2 \times 10^{-3}$ |
| | Exp 10 | 0.97 | 6 | 130 | 10 | 19 | 205 | 5.8 | $4.7 \times 10^{-4}$ |
| | Exp 11 | 0.97 | 5 | 144 | 10 | 22 | 228 | 5.9 | $3.5 \times 10^{-4}$ |

*values retrieved just before lights on

**values retrieved at integrated OH exposure $= 5 \times 10^6$ molecules $cm^{-3}$ hour

*Page 8, Lines 14-17: Please clarify (with evidence and explanation) that the nature of the artifacts that should have been associated with the prior studies based on methodology are actually consistent with the differences between the prior studies and the present study, as is vaguely suggested in the submitted manuscript.*

In this section, we highlight the differences observed with other studies regarding the contribution of anhydrosugars to the total POA mass concentration. We argue that the differences stem from the type of set-up used: collection on quartz fiber filter of the emissions in a dilution tunnel vs online collection and analysis by TAG-AMS in a smog chamber.

Different artefacts in each of the method can cause the observed discrepancy. Sampling artefacts with filters are relatively well known although complex. The duration of the sampling period, the temperature, the concentration of OA, the dilution ratio can influence the measured concentration. (Eatough et al., 1990) and (Turpin et al., 1994) have estimated 80 % of the mass collected on filters can be lost due to volatilization (negative artefacts) and up to 50 % of the mass can be added to due partitioning of the semi-volatile compounds (SVOCs) onto the surface of the filters (positive artefact). TAG-AMS is free from these sampling artefacts. Uncertainties remain however regarding the potential degradation of the analytes during thermo-desorption.

An especially relevant point here is the dilution ratio. Between the ejector diluter and injection into a smog chamber, we estimate at 200 the total dilution factor. This is nearly 10 times what (Fine et al., 2001) estimate for their own set-up. In accordance to the partitioning theory of Odum et al., (1996) it is expected at lower dilution that more SVOCs will partition to the particulate phase (Robinson et al., 2007) and onto the filters, thus increasing the overall OA mass and decreasing the observed contribution of primary compounds such as levoglucosan.

Nevertheless, we modified the text as follow:

The conditions and methods with which they sampled the emissions were however different (on quartz fibers filters) with a dilution factor at least 10 times less than what is used in this study. This could result in a higher fraction of SVOCs partitioning to the particulate phase, therefore increasing the overall OA mass, and thus decreasing the individual contribution of the markers.

*Page 8, Line 27: Methoxyphenols are not the 'predominant' class of species in the primary BBOA – Figure 3 clearly shows that anhydrosugars comprise the predominant fraction of the identifiable OA.*

We modified the text line 27, page 8 to:

The methoxyphenols account for an important fraction of the POA.

*Page 9, Lines 5-8: This sentence embodies an important criticism of the data interpretation within this study. The authors describe differences between the softwood and hardwood samples. At the same time, hardwoods and softwood fuels were burned under different conditions. The authors seem to interpret differences between the log wood stoves and the pellet stove as differences in fuel at this point, while at other points, the differences are attributed to burning conditions. If the interpretation given here is thought to be accurate, then an explanation for why burning conditions don't influence the observed difference must be described.*

We agree with reviewer #1 that our interpretation of the data was not conservative all throughout of the manuscript. Note, this comment was part of the general comment of the paper by anonymous reviewer 1. Therefore we refer the reader to this section of the review for our response.

*Section 3.2.1: The authors state that the primary compounds initially represent 48% of the POA, but after oxidation (4-6 hours) they represent < 8%... of what? Of POA? Of total OA? Can this just be described by the addition of SOA mass to the particles?*

After oxidation, the primary compounds represent < 8 % of the total OA mass concentration measured with HR-ToF-AMS. The decrease in their contribution is a combination of at least two main effects:

- Actual depletion (oxidation and/or vapor wall loss)
- Additional SOA mass

We modified line 29/30 page 10 as follow:

The identified primary compounds initially represent 48 % of the total POA mass concentration but < 8 % of the total OA mass concentration after 4 - 6 hours of atmospheric aging (bin 4).

*Figure 6 shows that the relative contribution of levoglucosan is somewhat stable around 40 % of identified OA throughout the experiment. Please clarify what is being quantified and to what fraction of the aerosol the percentage is referring.*

On Figure 6 we show that levoglucosan represent at any time during the experiment 30 – 40 % of the identified (with TAG) OA mass fraction. The contribution of levoglucosan to the total OA mass concentration measured by TAG-AMS however decreases over time as demonstrated in Figure 8.

We revised Figure 6 (labels and legend) in the manuscript in order to clarify this (Figure 4 here).

[Figure]

**Figure 4: Typical evolution of the chemical fingerprint of the organic aerosol emitted by biomass burning during atmospheric aging (Exp. 2, stove A). "Identified OA mass" refers to the OA mass concentration whose molecular composition is resolved by TAG-AMS.**

*It seems from this text that the mass concentration of levoglucosan, for example, decreased substantially during exposure to OH, but Figure 6 suggests that it is relatively stable against aging based on how the data are shown. Then again, Figure 7 shows the ER = 0.6, so the actual concentration is decreasing during aging. Figure 8 shows that levoglucosan is decreasing as a fraction of total OA. Basically, Figure 6 seems inconsistent with Figs 7 and 8. Is the change in the identifiable fraction playing a role? It is very likely that the data are somehow telling a consistent story, but it is not presented clearly. [Note that this comment explicitly references levoglucosan, but the comment also applies to other Primary Compounds.]*

As noted by anonymous reviewer #1, the figures show different but consistent evolutions regarding levoglucosan (and other markers). As the OA mass increases during aging and as the compounds are depleted, one has to carefully look at the referential to which the concentration of the marker is normalized to. Through figure 6 to 8 we choose 3 different referential. Each one aims at illustrating one specific aspect of the modification of the chemical fingerprint during aging.

In Figure 6 of the manuscript, we show a typical and comprehensive evolution (through one example) of the different parameters measured within this study (top panel : evolution of the OA mass concentration measured by the HR-ToF-AMS, 2nd panel : the fraction of total OA quantified/identifed by TAG, 3rd panel : total absolute concentration of the compounds quantified by TAG - which remains sratherconstant all throughout the aging - and bottom panel : relative contributions of all the compounds quantified by TAG to the identified OA fraction). We agree that showing the relative concentration of the organic markers to the total OA mass concentration would be more pertinent, but the very fast drop of the total identified OA prevents such representation. For better visibility, we choose to represent the contribution of organic markers to the total identified OA. As the total absolute concentration of the identified fraction (3rd panel) is rather constant all throughout the experiment, this plot provides a good comparison of the concentration of the secondary markers with that of the primary markers, during the aging process.

One would have to refer to Figure 8 (top panel) to see the evolution of the contribution of levoglucosan to the total OA mass concentration measured by HR-ToF-AMS. There, the contribution of levoglucosan is shown to decrease as SOA is formed and levoglucosan depletes (due to oxidation and/or vapor wall loss ). This information is especially important in the context of source apportionment studies (eg. Chemical Mass Balance model assumes the contribution of the markers to total OA is constant over time).

Finally, in Figure 7, we look at the enhancement ratio. A compound with an ER > 1 would indicate the compound is formed during aging. An ER < 1 relates to the decay of a compound (i.e. if ER = 0.2, 80 % of the mass of the compound is lost during aging). The study of the ER of each of the marker allows us to differentiate between primary and secondary type of compounds.

*Page12,Line33–Page13,Line2: What is the relevance of the ratios of nitrocatechols to levoglucosan, since it is known that the latter is consumed during OH exposure?*

*Page 13, Line 3: Higher levels of 4-NC are discussed with reference to prior observations, however far larger discrepancies between the prior observations and the present study exist for MNC/LG. The 4-NC/LG ratio range at least overlaps the prior studies, but the MNC/LG ratios are shifted by about a factor of 10, yet are not discussed.*

The ratios of nitrocatechols to levoglucosan in the smog chamber vs ambient put into perspective the concentration of the secondary markers formed during our experiments.

The ratios NC/LG and MNC/LG for these experiments indicated in the text have been switched. We apologize for this mistake. The 4-NC/LG ratio observed in this study are 0.25 – 0.5 while the MNC/LG ratios vary from 0.06 to 0.15. As such, the ratio of MNC/LG largely overlaps with what is observed in the ambient while the NC/LG is much higher as mentioned later in the text.

As explained, we hypothesize the high level of 4-NC could be related to the high level of NOx also present in the chamber.

We thank anonymous reviewer #1 for his comment and have corrected the text line 3, page 13 as follow.

Here, the 4-NC/LG and MNC/LG ratios vary from 0.25 to 0.5 and 0.06 to 0.15, respectively.

*Page 13, Line 7: Does "the mass spectra (EI, 70 eV)" refer to those obtained from a database? Please clarify. If from a database, please cite the reference.*

The 4-nitrocatechols mass spectrum was obtained from the NIST database. We have included this reference in the text. The methyl-nitrocatechols mass spectrum was obtained from in-lab analysis of 3-Methyl-5-Nitrocatechol by GC/MS (Thermo Trace GC 2000-Polaris Q) (without derivatization).

*Page 13, Line 23-24: The impact of burning conditions is less clear since the fuel was different for Stove C. Please clarify this issue in all instances within the manuscript, including Page 9, Lines 5-8 as discussed in a comment above.*

We revised this section of our conclusions in the following manner:

The emissions factors of the individual organic markers are mostly driven by the MCE. Smoldering combustion increases the EF of the reported compounds. Within the experiments conducted with the same fuel, the contribution of the markers to the total POA mass concentration also vary according to the MCE (i.e. lower contribution at smoldering conditions). This indicates that smoldering combustion induces the emission of OA with a more complex composition. The contribution of levoglucosan for instance varies from 40 % – 50 % at the highest MCE, down to 15 % at the lowest MCE.

*Page 14, Lines 1-3: How might measurements be improved to detect and quantify a greater fraction of the POA and SOA mass? Can this be inferred from the limitations of TAG-AMS? [A discussion of this topic may not be entirely appropriate for the conclusion section, but may be a worthwhile topic to include elsewhere in the manuscript.]*

TAG-AMS, like most GC/MS is limited in its capacity at eluting highly oxygenated and high molecular weight species. While it is understood that the detectable OA fraction by the TAG-AMS could potentially reach 100 % in the case of an aerosol purely composed of hydrocarbons (Williams et al., 2015) the instrument typically only detects around 20 % of the total OA (Williams et al., 2006) in an ambient set-up.

A first approach to circumvent this limitation is to focus on the thermal decomposition window i.e. the signal output by TAG during thermal desorption of the analytes (300 °C – 310 °C). Fortenberry et al., (2017) and Williams et al. (2015) have observed a signal increase for specific ions which they relate to the decomposition of high molecular weight molecules and thermally labile oxygenated OA. While this method does not provide information at the molecular level, it gives insights into the overall oxidation level of the bulk OA sample and can provide time series of fragments which could be used as tracer for fresh and aged BBOA.

However, the version of the TAG-AMS described here could not be operated in this way.

*Technical corrections – Page 3, Line 4: correct the in-line citation Page 9, Line 3: correct the in-line citation Page 9, Line 21: consider beginning a new paragraph for fatty acids Page 13, Line 8: "Figure 10" should be "Figure 9"*

Corrected as suggested.

[revised manuscript text omitted]

---

## Author Comment (AC2) · 26 Apr 2018

We thank the Referee for the careful revision and comments which helped to improve the overall quality of the manuscript. A point-by-point answer (in regular typeset) to the referee's remarks (in the *italic typeset*) follows, while changes to the manuscript are indicated in blue font. In the following document, lines references refer to the manuscript version reviewed by the anonymous referee.

**Anonymous Referee #2

*This paper uses TAG-AMS to quantify compounds in organic aerosol from two logstoves and one pellet stove after emission and after equivalent atmospheric aging in a chamber. Three categories of compounds are described, and the emissions factors and emission profiles of biomass burning tracers are measured. The authors quantify how the OA changes after atmospheric aging. This paper is generally well-written but would benefit from a read through for typos (see examples in editorial section). Additionally, more discussion on implications of this study could be included in the conclusions.*

*Some suggestions for improvement are below, but overall the paper is interesting and worthy of publication.*

*General.*

*The variability in the total OA and the resolved OA for each stove type is interesting and should be further explained. It is unclear that the three or four samples from each should be simply averaged as was done in Figure 3. That resulted in one composition for each stove type without anything showing the potential variability in the composition for each stove. The discussion in the text regarding this composition should clarify if it is in regard to the total POA or the resolved fraction of POA. This should be stated clearly and be used consistently.*

Regarding the initial POA concentrations injected in the chamber, we tried as much as possible to inject low, medium and high concentrations but always representative of atmospheric conditions (from ambient to plume like concentrations). The variability of POA emissions factors and secondary aerosol production potential is fully described in Bertrand et al., (2017). As for the organic markers, the main driver of this variability is the MCE. The section 3.1 has been significantly rewritten to clarify our discussion and conclusions according to anonymous reviewer #1's second general comment and the present comment. Please refer to our reply to anonymous reviewer 1. Figure 3 has also been changed accordingly (see figure 3 of this document).

*Specific comments are below.*

*In the section on atmospheric aging, the authors should be more explicit about what measurements went into the results of the gaining experiment. It seems as though all of the OA was aged, but there is no explicit mention of which stove was used to generate the POA that was then aged. The initial composition of the POA could influence the SOA composition, so there should be more discussion of this, unless the resolved fraction is too small to draw conclusions.*

Primary emissions from all three stoves were aged. As suggested by reviewer #2, it is possible that the initial composition could influence SOA composition. However, we do not observe significant differences between the experiments regarding the individual evolution of each of the marker during aging. Therefore, we report here average values derived from all the experiments regardless of the type of stove. With such a large data set, and many possible entries, we believe it was better to present a global overview of the most important results and to focus on the main drivers of the variability. Nevertheless, as we are aware of the relevance of such a large data set

for the scientific community, we have made all the results, of each experiment, available in the supporting information.

Nevertheless, we modified the manuscript line4, page 10 to include the following section:

Note, while it is possible differences in the composition of the primary emissions can influence the SOA composition, we have not found significant differences in the evolution of the individual markers. In this section, we therefore consider all aging experiments from the three stoves. EFs of the molecular markers at an integrated OH exposure of $5 \times 10^6$ molecules cm$^{-3}$ h for each experiment are reported in Table S3.

*P. 3, L. 6-9: It would be better to incorporate these references into the paragraph and the following sentences. More detail is included later about the biomass burning, but not the other sources. P. 3, L. 17: This is redundant with the previous mention of Levoglucosan at L. 11. Remove or rephrase.*

As this publication is limited to the study of biomass burning, we choose not to detail the other sources. As per the suggestion of anonymous reviewer 1 and 2 we have nonetheless reworked the introduction (see above).

*P. 4 – Is particle bounce a factor in the TAG-AMS? Was that taken into account in the calculations of EFs? How does the TAG-AMS treat organics on black carbon? Can the black carbon be detected?*

A humidifier on the sampling line is used to eliminate particle bounce during the impaction of the particles in the collection cell of the TAG-AMS. As particle density decreases with water uptakes, this creates a compensating effect. Therefore the aerodynamic diameter remains essentially unchanged (Williams et al. 2006). As such we do not need to correct for the particle bounce in our calculation of the concentration and EF.

The temperature (280 °C – 300 °C) at which we operate the TAG-AMS does not allow for the analysis of black carbon. As stated in Isaacman et al. (2014) eBC is essentially treated as a "non-volatile contaminant" for TAG-AMS that requires regular cleaning of the collection cell.

As it is there is no conclusive proof showing either the adsorbed organics will come off the black carbon during thermal desorption or if the adsorbed organics are permanently stuck on the black carbon.

It is worth nothing however that the amount of black carbon is only a very small fraction of the PM measured during most of these experiments (Stove A and B).

*P. 4, L. 30: Comment on what is causing such a large range in OA concentration. Does this have anything to do with the amount of time it was burning? How was this controlled for? Does the*

*emission factor of each stove change with the amount of time the fuel has been burning? How did the emissions change over the 30 minutes they were stabilizing in the chamber? More details should be added here.*

We injected the emission past the startup operation at once regardless of the efficiency of the combustion (flaming vs smoldering). Typically smoldering type of fire required a shorter injection time before the OA concentration observed was no more relevant to what can be observed in the atmosphere. In the analysis of the data that followed, we determined the large range in OA concentration was directly related to the combustion efficiency.

The pellet stove was entirely automated, hence the little to no variability in the type of combustion (pure flaming) and POA concentration. The logwood stoves however are manually operated and overall harder to control. The MCE was found to vary on a large range during these experiments, so did the OA. The initial stabilization was determined not to influence the relative composition of the primary emission.

More details on the combustion conditions can be found in a previous publication (Bertrand et al., 2017) that investigates the effect of the combustion condition on the POA EF and Secondary Organic Aerosol Production Potential (SAPP) and OA chemical fingerprint by AMS.

We modified the text line 2, page 8 to better highlight this reference.

More details regarding the burn variability can be found in Bertrand et al. (2017).

*Table 1: What is the difference between "in the chamber before lights-on" and "values retrieved just before lights on"? What is the timing difference between the [BC] and the [NOX]? Why are the concentrations of BC, etc. not consistent? What other conditions were changed?*

"Before lights on" is the more generic term to refer to the primary emissions. "Just before lights-on" should indicate to the reader that all EF and ER are based on the concentration of the OA and marker retrieved after a homogenization and stabilization period.

There is no timing difference between the [BC] and the [NOX]. The NOx concentration was also retrieved just before lights-on. We thank anonymous reviewer # 2 for his comment. We have modified Table 1 accordingly.

We assume that by "not consistent", anonymous reviewer #2 is referring to the large gap between the concentration in eBC of Stove A/B and Stove C. The pellet stove (Stove C) was found to emit a large amount of eBC. The OC/eBC ratio for each of the stove is as follow: 5 for stove A, 3 for stove B, 0.05 for Stove C. We determined the OC/eBC of our stove is much lower to what is found in the literature (0.9 – 4). As discussed in a previous publication (Bertrand et al., 2017), data in the literature regarding this type of appliances and with similar power outputs are scarce and therefore we remain unable to conclude whether this result is a singularity of our stove or if it could be reproduced at a larger scale with other similar equipment. Here, we hypothesize the

stove design and fuel loading technique contribute to high temperature and fuel-rich zone in the combustion chamber, thus increasing the level of eBC emitted.

*Figure 1: The words are all very small. It might be possible to shift so that the figure takes up less space (i.e. is less diagonal).*

We revised Figure 1 as suggested by anonymous reviewer 2 (Figure 1).

[Figure]

**Figure 1. Schemes of the atmospheric chamber set-up (a) and TAG-AMS with in-situ derivatization (b).**

*P. 5, L. 16: "discrimination of all fragments below m/z 10" – Explain or rephrase.*
We reword the sentence in the following manner:

The quadrupole deflects any ions at m/z < 10.

*P. 6, L. 10: Wall losses decreased the concentration of aerosol, and this was accounted for qualitatively by increasing the sampling length as time progressed. Mention here that the quantitative estimates for wall losses are included in Section 2.4. These questions came up here before reaching Section 2.4: Are there any calculations or quantitative estimates of the wall losses? How much time was added to each sampling period, and how was that calculated?*

As suggested by anonymous reviewer #2 we modified the text page 6 line 10 to include the reference to section 2.4. As explained in this section, particle wall losses can be estimated based on the decrease of eBC. Vapor wall loss cannot be easily inferred from direct observation in the chamber. Therefore most studies do not account for this loss. It is assumed that the SOA formed thus corresponds to a lower estimate.

How the sampling time was varied can be seen on Figure 2. The time increase was a simple estimate based on a-priori knowledge of the detection limits of TAG-AMS and previous observation of the particle wall losses with that same chamber (Platt et al., 2013; Klein et al., 2016)

*Figure 2: The legend is somewhat confusing. Suggest removing "TAG-AMS Sampling" and changing the caption to be "Figure 2: Example of the TAG-AMS (light shading) and..."*

This issue was also raised by anonymous reviewer #1. As shown in Figure 1, we revised the colors and the legend of the figure.

*P. 6, L. 11: Was this an automated matching process? What confirmed a match? Comment on how were the initial standards and NIST spectra were selected for comparison.*

Standards and NIST spectra were selected based on a-priori knowledge. Identification of the compounds was not automated and was carried out by the author. A compound is identified if the retention time is in agreement with what is known of the chromatographic conditions, and if the mass spectrum correlates well with the mass spectrum of a standard and/or the NIST data base (R2 > 0.7). Identification has also been confirmed by 2D-GC-MS analysis.

*P. 6, L. 30: What is eBC? Write out or explain the first time it is used. On the next page, it is listed as black carbon. In equation 2, it is written as just "BC". Make sure these are consistent, and if they are different, explain what that difference is.*

The black carbon measured with an aethalometer is not considered 100 % black carbon. Therefore Petzold et al., (2013) recommend authors use the term "equivalent BC" (eBC) instead of BC. We have replaced the term BC with eBC all throughout the manuscript.

*Equation 1: Remove the "." – not standard formatting for equations.*

Equation 1 was corrected as suggested.

*P. 6 – Is there any preferential wall loss based on composition? The only aerosol tested is eBC. Is that representative of all of the aerosol?*

Even at longer wavelength, organic condensation on eBC can increase the measured absorption (Bruns et al., 2015). In smog chamber experiment this can be a problem especially minutes after lights-on when most of the SOA is formed. This should result in a modification of the signal which we do not observe on our data set.

Nevertheless, it is possible to retrieve the particle wall loss rate by fitting the decay in number concentrations measured by SMPS rather than using eBC (Weitkamp et al., 2007). Here both methods were carried out. They yielded similar results.

*P. 7, L. 7: How reasonable is that assumption? Was it tested?*

Anonymous reviewer #2 refers to this sentence: This method [for calculating particle wall loss] assumes that the condensable material partitions only to the suspended particles and vapor wall losses are considered negligible."

Unlike particle wall loss, vapor wall loss cannot easily be inferred from our observations. Estimating vapor wall losses require we first determine different parameters including the saturation vapor concentration, particle mass accommodation coefficient, or equivalent organic mass concentration at the wall. These parameters have yet to be well constrained and large uncertainties remain. Therefore, while the assumption to consider vapor wall losses negligible is not true, this is inevitable in smog chamber studies if one wants to nonetheless estimate a lower bound SAPP. Recently, several studies have attempted to retrieve the loss rate of SVOCs and the impact on the SOA formed (Matsunaga and Ziemann ‡, 2010; Zhang et al., 2015; La et al., 2016; Trump et al., 2016). Zhang et al. (2015) estimated that the SOA formed may actually be underestimated by a factor of 4 due to this assumption.

For more discussion regarding the effect of vapor wall losses on the markers, we refer anonymous reviewer #1 to a paper now in review in ACP. (Bertrand et al., 2018b).

*P. 7, L. 19: Clarify or rephrase the discussion of the symbols in Equation 2. "Here refers to the concentrations" – but really, it refers to the change in the mass of each.*

We modified page 7, line 19 as suggested by anonymous reviewer #2. The text is now as follow:

Here Δ refers to the change in concentration of the species in the atmospheric chamber between background and emission/stabilization.

*Figure 3: Reorganize the legend so that they are left justified. (If this is based on the groups discussed in the text, then include headings for each type). This figure would also be more useful if it could also demonstrate the fraction of total OA that it represents. It would also be nice to have some error bars or some representation of uncertainty. The fact that for Stove B, 26 to 85% of the OA was resolved at any time could also mean that there is large variability in the relative fractions. It appears that the main point of the graph is that Stove A and Stove B have similar compositions. But it would be good to see some representation of the differences in their total OA concentrations and the amount of OA resolved for each. In addition to or instead of these average bar plots, I suggest including a figure with a bar for each sample, grouped by stove type, with the y-axis representing the fraction of total OA resolved, instead.*

As suggested by both anonymous reviewers #1 and 2, we modified this figure to show the contribution of the identified OA to the total POA mass concentration for all experiments (Figure 3 here).

*P. 7, L. 28: Explain where the 26-85% of the number for the 26-85% of the mass comes from. In Figure 3, are the ranges due to averages taken for each sample? That is a lot of variability. That variability is in itself interesting and should be discussed.*

26 – 85 % refers to the fraction of total POA (measured by the HR-ToF-AMS) that the TAG was able to identify for each replicate.

As mentioned in section 3.1, we determined this large variation is mainly related to the combustion efficiency. Emissions with a higher MCE show a larger fraction of levoglucosan (the main compound of biomass burning emission), and typically a higher identified fraction (Figure 4). This indicates that the emissions from a burn with a better combustion efficiency (high MCE) are less complex (contains mainly levoglucosan and methoxyphenols) that emissions resulting from burns with a lesser combustion efficiency (low MCE). In the latter case, we remain unable to identify a large majority of the compounds emitted most probably due to the emissions of high MW oxygenated compounds.

*P. 8, L. 4: This statement is not consistent with the figure. There is huge variability in the fractions of OA identified in all three stoves. Stove B has the lowest low range but also highest high range, so it is an overgeneralization to say that Stoves A and B are less resolved. Or, more data and discussion needs to be added to support that.*

We do agree that there exists an important variability in the fraction of OA identified which reflects the very high variability of any biomass burning experiments. Again, we strongly believe that the consideration of the MCE significantly reduce this variability (but obviously not totally). Here we meant to show that the resolved fraction can partly be explained by combustion efficiency, regardless of which stoves we look at. Nevertheless we removed this statement from the revised manuscript.

*Figure 4: Remove the gap between a/b and c/d and the x-axis labels, since they are the same as those at the bottom for e/f. Why are R values on c/d and not on the others? Remove them or add the others to be consistent.*

Figure 4 has been modified as follows:

[Figure]

**Figure 2. Emissions factors of POA, levoglucosan, and methoxyphenols, as well as their contribution to the total OA as a function of the modified combustion efficiency (MCE). Details on how the MCE was calculated are given in Bertrand et al. (2017).**

*P. 8 – Are these fractions of total POA or resolved POA? There is a huge range in total POA as well as resolved POA, and there appears to be a range in these sugars as well. If the standard deviations are included in the comparison, there is no statistical difference in the percentages of anhydrosugars from the three stoves.*

These fractions refer to the total POA mass concentration (as measured by HR-ToF-AMS). Following anonymous reviewer #2's observations of the lack of statistical differences between the stoves regarding the contributions of anydrosugars to the total POA mass concentration, we removed this comment from the manuscript. We revised the text as follow:

The EFs of levoglucosan are in the range 26 - 249 mg kg$^{-1}$ of fuel or 13 - 45 % of the total POA mass concentration. Levoglucosan is emitted along with its two isomers, mannosan, and galactosan. The EFs of mannosan are 4 - 20 mg kg$^{-1}$ and represent an average fraction of the POA mass concentration of < 4 % ± 2 % ($n$ = 11). The EFs of galactosan are 0.5 - 10 mg kg$^{-1}$ and contribute < 1 % of the POA mass concentration. Therefore, anhydrosugars represent 27 – 52 % of the total POA mass concentration

*P. 9, L. 27: It is unclear how Figure 5 illustrates that.*
*Figure 5: State the difference between a and b in the caption. This figure could also be moved to the supplement. It is only referenced once.*

Figure 5 highlights the compounds which are formed or removed from the particulate phase during an aging experiment. We do agree that this figure is not fundamental for the paper, but we truly believe that this figure is very didactic and highlights very well the potential changes of the molecular composition of biomass burning aerosol once injected in the atmosphere. The figure should in our opinion be left in the main text for its pedagogical virtues. Such figures are scarce in scientific papers. Nevertheless we revised the introduction of the figure in the revised version of the manuscript p9, line 27:

The molecular fingerprint of the BBOA undergoes important modifications during aging. By highlighting the differences observed between a sample of fresh and aged biomass burning emissions, Figure 5 illustrates well the impact of such aging on the molecular fingerprint of the emissions.

We also modified the caption of the figure as follows:
Figure 5: Overview of the change occurring in the chemical fingerprint at the molecular level of biomass burning organic aerosol during atmospheric aging. Organic markers decaying (green) or forming (red) during the aging. The figure is a three-dimensional representation of the difference in intensity of the peaks (normalized to the signal of the internal standards) calculated between aged (integrated OH exposure of $5 \times 10^6$ molecules cm$^{-3}$ h) and fresh samples. The samples were collected on quartz fiber filters and analyzed with 2D-GC VUV/EI (see SI). (a) view from the top, (b) view from the bottom.

*Table 2: Remove "Bin" labels, or explain them. It seems like enough for each column to have integrated OH exposure. Is this for one sample of one of the stoves or an average? It is unclear what experimental set up this data came from.*

It is in our opinion that Bin labels should remain to serve as a reference to Figure 6.
As the contribution of the markers to the total OA mass concentration does not vary significantly between the type of stove, we report in Table 2 contributions averaged from all experiments (all the results for bin 0 and 5 for each experiment are reported in the SI). For clarification, we modified the legend as follow:

pWLC contribution of the compounds to the total OA mass concentration at different times of the photo-oxidative process. This table only contains a list of selected compounds and average values. Data from all experiments are divided into seven bins of different integrated OH exposure intervals (from 0 to $9 \times 10^6$ molecules cm$^{-3}$ hour) then averaged. A full list, with minimum and maximum values is also available in the SI.

*P. 10 – Clarify the units of OH exposure. This may take a new reader some time to interpret. The "integrated OH exposure" seems to be just the OH concentration (molecules/cm3) multiplied by the time exposed (hours).*

As suggested by anonymous reviewer #2, we modified the part of the section 2.2 about OH exposure to include this information:

We retrieved the integrated OH exposure (OH concentration integrated over time) based on the differential reactivity of butanol-D9 and naphthalene

*Figure 6: Same legend comment as Figure 3. What POA was used in this figure? What stove is this from?*

Identified OA refers to the fraction of total OA (measured with HR-ToF-AMS) that is identified by TAG-AMS. As this clarification has also been asked by anonymous reviewer #1, we have taken care to explicit whether OA referred to total OA or identified OA in every section of the manuscript where it was necessary. In addition, we have added page 4 line 10 - 15 a brief definition of total OA and identified OA.

We modified the legend of the figure as follow:

Typical evolution of the chemical fingerprint of the organic aerosol emitted by biomass burning during atmospheric aging (Exp. 2, stove A). "Identified OA mass" refers to the OA mass concentration whose molecular composition is resolved by TAG-AMS.

*P 10 – Why does the fraction of identified OA decrease with OH exposure? This almost implies an inverse correlation between OA mass and fraction of identified OA. It seems like the OA produced at the very end is then all unidentifiable.*

Anonymous reviewer #2 is right, the OA produced at the very end is quasi-unidentifiable with this method. Two effects are at play here which can explain this.

The most important effect is the change in composition of the OA with aging. The overall volatility of the OA decreases with aging as the O/C ratio increases (Bertrand et al, 2017). TAG-AMS is known (Williams et al., 2006, 2015) to have a lower capability of eluting the highly oxygenated type of compounds that constitute the SOA fraction, hence why as aging goes on, the fraction of identified OA by TAG-AMS decreases.

In addition, while the total OA increases, most of the identified markers are also primary compounds, whose concentration decreases because of oxidation and/or vapor wall losses resulting in their contribution being less during aging than it was in the primary emissions and thus further contributing to the overall decrease of the identified OA fraction For instance, the concentration of levoglucosan whose contribution to the POA total mass concentration averaged at 30 % drops by about 40 % during aging, resulting in a contribution less than 5 % to the total OA mass concentration.

*Figure 7: This figure is interesting, and the split between the red and blue is nice. But having the primary compounds with upside down bars make them seem negative. Additionally, the ER ratio increases from left to right, but the bars decrease in size, which is counterintuitive. The caption should state what the error bars are calculated from (and the text could comment on why they vary significantly).*

We believe that by having the primary compounds upside down, the reader can quickly discern that these compounds have a different behavior than the others (their absolute concentration decays during aging). Furthermore, we use here the size of the bar as an indication of the importance of the decay, in a similar way that for the secondary compounds the size of the bar indicates how much compounds form.

Error bars were calculated from the standard deviation of the different enhancements derived from all the experiments. Their variability can result of several parameters, inherent variability of the aging, slight differences in the conditions of the primary emissions, signal output for some of the compounds near the detection limit. This variability however does not change the perception of the compounds whether they are primary, non-conventional primary, or secondary.

We revised the legend of the Figure:

[Figure]

**Figure 7. Mean enhancement ratios (ER) of individual compounds in biomass burning emissions. The ER is defined as the pWLC concentration measured in the particle phase within a time range corresponding to an integrated OH exposure of $4 \times 10^6$ - $6 \times 10^6$ molecules cm$^{-3}$ hour (bin 4, Figure 6) and divided by the concentration in the particle phase measured at $t_0$ (ie. before lights on). An ER of 0.8 means a loss of 20 % during aging after pWLC. Note that for the secondary compounds (colored in red) observed after the start of the photo-oxidative process only, the ER are calculated using the detection limit. Error bars show the standard deviation derived from all experiments.**

*P. 11, L. 19: Interesting discussion!*

*Figure 9: In the text, this is listed as Figure 10. What is the explanation for the difference between the two measurement types? It would be good to show the slope here, in addition to the R squared value.*

The figure is now numbered as Figure 9 in the revised version of the manuscript. Intensities of $C_6H_5NO_4^+$ (m/z 155.022) and $C_6H_2NO_4^+$ (m/z 151.098) from the HR-ToF-AMS data are expressed, here, in µg m$^{-3}$ nitrate equivalent which reflects the peak intensities rather than an actual concentration. A dedicated calibration of the HR-ToF-AMS is needed for these compounds to be quantitative. This figure highlights the linear relationships existing between those 2 molecular ions intensities with the actual concentrations of nitrocatechols measured by the TAG and do not constitute an intercomparison of measurement methods. Those linear relationships can

eventually be used as calibrations curves in order to provide of rough estimation of the nitrocatechols concentration in future studies. Discrepancies can be due to interferences with the signal of other compounds with similar fragments than the nitrocatechols ones, and, in the case of TAG measurements, analytical uncertainties (15 – 20 %).

As suggested by anonymous reviewer #2 we have modified Figure 9 to include the slope (Figure 3). The caption has also been modified.

[Figure]

**Figure 3: HR-ToF-AMS measurements against TAG-AMS measurements. $C_6H_5NO_4^+$ (*m/z* 155.022) vs. 4-nitrocatechol (top), and $C_6H_2NO_4^+$ (*m/z* 151.098) vs. alkylated nitrocatechols (3-methyl-5-nitrocatechol and 4-methyl-5-nitrocatechol) (bottom), during the aging of biomass burning emissions. Intensities of $C_6H_5NO_4^+$ (m/z 155.022) and $C_6H_2NO_4^+$ (m/z 151.098) from the HR-ToF-AMS data are expressed here in µg m-3 nitrate equivalent which reflects the peak intensities and not an actual concentration. A dedicated calibration of the HR-ToF-AMS is needed.**

*P. 14 – Add a couple more sentences about the implications of these results and why it is important that they were measured.*

We included the following aspects in the revised version of our manuscript:

These data will serve to improve our ability to properly apportion the biomass burning source in the ambient atmosphere. Already, source apportionment studies reveal a large fraction of the aged OA originate from modern sources (i.e. non-fossil fuel sources). The secondary compounds highlighted in this study thus should be evaluated to serve as potential tracers of the aged biomass burning emissions. The lack of stability of the primary compounds in the context of source apportionment study and how this might result in an underestimated contribution of BBOA should also be further examined.

*Editorial: P. 2, L. 3: Add "of" after "evolution"*
*P. 2, L. 6: Remove redundant "emission"*
*P. 3, L. 21: Change "during" to "due to"*
*P. 8, L. 19: Figure 6? Should this be Figure 4?*
*P. 10, L. 23: Add "with" after "along"*
*P. 13, L. 8: Change Figure 10 to Figure 9*

Corrected as suggested.

[revised manuscript text omitted]